# TRIP13 is a protein-remodeling AAA+ ATPase that catalyzes MAD2 conformation switching

Qiaozhen Ye[1], Scott C Rosenberg[1], Arne Moeller[2], Jeffrey A Speir[2], Tiffany Y Su[1], Kevin D Corbett[1,3]*

[1]Ludwig Institute for Cancer Research, San Diego Branch, La Jolla, United States; [2]National Resource for Automated Molecular Microscopy, Department of Integrative Structural and Computational Biology, The Scripps Research Institute, La Jolla, United States; [3]Department of Cellular and Molecular Medicine, University of California, San Diego, La Jolla, United States

**Abstract** The AAA+ family ATPase TRIP13 is a key regulator of meiotic recombination and the spindle assembly checkpoint, acting on signaling proteins of the conserved HORMA domain family. Here we present the structure of the *Caenorhabditis elegans* TRIP13 ortholog PCH-2, revealing a new family of AAA+ ATPase protein remodelers. PCH-2 possesses a substrate-recognition domain related to those of the protein remodelers NSF and p97, while its overall hexameric architecture and likely structural mechanism bear close similarities to the bacterial protein unfoldase ClpX. We find that TRIP13, aided by the adapter protein p31(comet), converts the HORMA-family spindle checkpoint protein MAD2 from a signaling-active 'closed' conformer to an inactive 'open' conformer. We propose that TRIP13 and p31(comet) collaborate to inactivate the spindle assembly checkpoint through MAD2 conformational conversion and disassembly of mitotic checkpoint complexes. A parallel HORMA protein disassembly activity likely underlies TRIP13's critical regulatory functions in meiotic chromosome structure and recombination.

*For correspondence: kcorbett@ucsd.edu

**Competing interests:** The authors declare that no competing interests exist.

## Introduction

The assembly and disassembly of specific protein complexes underlies many important signaling pathways in the cell. The HORMA domain (*Aravind and Koonin, 1998*) is a conserved, structurally unique signaling module that forms complexes through a characteristic 'safety-belt' interaction in which the C-terminus of the domain wraps entirely around a short region of a binding partner (*Luo et al., 2002*; *Sironi et al., 2002*; *Hara et al., 2010*; *Kim et al., 2014*). HORMA domain protein complexes participate in multiple cellular signaling pathways, including meiotic recombination control, DNA repair, and the spindle assembly checkpoint (SAC). While the regulated assembly of HORMA domain protein complexes has been extensively characterized, the mechanisms underlying their disassembly, which requires significant conformational changes to disrupt the extremely stable safety-belt interaction, are largely unknown.

In meiosis, homologous chromosomes must recognize one another and recombine, forming physical links called crossovers (COs) that enable their bi-orientation and segregation in meiosis I (*Zickler and Kleckner, 1999*). CO formation is promoted and regulated by a conserved family of HORMA domain proteins termed HORMADs. Early in meiotic prophase, HORMADs localize to chromosomes along their entire lengths, where they promote the introduction of DNA double-strand breaks and bias the repair of those breaks toward the homologous chromosome (*Subramanian and Hochwagen, 2014*). In both yeast and mammals, excess recombination is limited by a feedback

**eLife digest** The genetic material inside human and other animal cells is made of DNA and is packaged in structures called chromosomes. Before a cell divides, the entire set of chromosomes is copied so that each chromosome is now made of two identical sister 'chromatids'.

Next, the chromosomes line up on a structure called the spindle, which is made of filaments called microtubules. Cells have a surveillance system known as the spindle assembly checkpoint that halts cell division until every chromosome is correctly aligned on the spindle. Once the chromosomes are in place, the checkpoint is turned off and the spindle pulls the chromatids apart so that each daughter cell receives a complete set of chromosomes.

A protein called MAD2 plays an important role in the spindle assembly checkpoint. It can adopt two distinct shapes: in the 'closed' shape it is active and halts cell division, but in the 'open' shape it is inactive and allows cell division to proceed. Another protein called TRIP13 can help turn off the checkpoint, but it is not clear how this works or whether TRIP13 acts on MAD2 directly.

Here, Ye et al. studied these proteins using a technique called X-ray crystallography and several biochemical techniques. The experiments show that TRIP13 belongs to a family of proteins known as 'AAA-ATPases', which can unfold proteins to alter their activity. Ye et al. found that TRIP13 binds to an adaptor protein that allows it to bind to the closed form of MAD2. TRIP13 then unfolds a part of the MAD2 protein, converting MAD2 into the open shape.

Ye et al. propose that, once all chromosomes are lined up on the spindle, TRIP13 turns off the spindle assembly checkpoint by converting closed MAD2 to open MAD2. Also, when cells are not undergoing cell division, TRIP13 may maintain MAD2 in the open shape to prevent cells from turning on the spindle assembly checkpoint at the wrong time. Further work will be needed to show how TRIP13 recognizes the closed form of MAD2, and whether it can act in a similar way on other proteins in the cell.

mechanism that removes or redistributes HORMADs along the chromosome after sufficient COs have formed. The removal of HORMADs depends on a conserved AAA+ family ATPase, Pch2/TRIP13, without which the frequency and spatial distribution of COs is disrupted (*San-Segundo and Roeder, 1999*; *Börner et al., 2008*; *Joshi et al., 2009*; *Wojtasz et al., 2009*; *Roig et al., 2010*; *Chen et al., 2014*). We have previously shown that the HORMADs assemble into higher-order oligomers through head-to-tail safety-belt interactions, and that these interactions are critical for their meiotic functions (*Kim et al., 2014*). As a predominant family of AAA+ ATPases function to disaggregate or disassemble protein complexes (*Erzberger and Berger, 2006*; *Sauer and Baker, 2011*), it has been proposed that Pch2/TRIP13 mediates HORMAD removal from chromosomes through specific recognition and disassembly of chromosome-associated HORMAD complexes (*Chen et al., 2014*).

Recently, mammalian TRIP13 has been shown to regulate the SAC, which monitors kinetochore-microtubule attachment in both mitosis and meiosis (*Musacchio and Salmon, 2007*). In this pathway, unattached kinetochores generate an inhibitor of the anaphase promoting complex/cyclosome (APC/C) called the mitotic checkpoint complex (MCC), which is composed of the MAD2, CDC20, BUBR1, and BUB3 proteins (*Hardwick et al., 2000*; *Fraschini et al., 2001*; *Sudakin et al., 2001*). MAD2 is a relative of the meiotic HORMADs, and exists in one of two conformers: an inactive 'open' state (O-MAD2), and an active 'closed' state (C-MAD2) (*Figure 8—figure supplement 1A*) that binds CDC20 through a safety-belt interaction to form the core of the MCC (*Sironi et al., 2002*; *Luo et al., 2004*; *Shah et al., 2004*; *Chao et al., 2012*). After all kinetochores become properly attached to microtubules, new MCC assembly is halted and the SAC is inactivated. Timely SAC inactivation requires two factors, TRIP13 (*Wang et al., 2014*) and the HORMA domain protein p31(comet) (*Habu et al., 2002*; *Xia et al., 2004*; *Hagan et al., 2011*; *Varetti et al., 2011*; *Westhorpe et al., 2011*; *Ma et al., 2012*), which recent evidence suggests may act together to directly disassemble the MCC. p31(comet) specifically recognizes and binds C-MAD2, and the p31(comet)-MAD2 interface overlaps MAD2's interface with BUBR1 in the intact MCC (*Xia et al., 2004*; *Yang et al., 2007*; *Tipton et al., 2011b*; *Chao et al., 2012*), suggesting that p31(comet) may compete with BUBR1 for MAD2 binding. Further, the combined activities of p31(comet) and TRIP13 can cause the dissociation of MAD2 from immunoprecipitated CDC20 or BUBR1 complexes in vitro (*Teichner et al., 2011*; *Eytan et al., 2014*).

Intriguingly, human *TRIP13* has also been identified as an oncogene: *TRIP13* is overexpressed in a number of human cancers (*Larkin et al., 2012*; *van Kester et al., 2012*; *Banerjee et al., 2014*; *Wang et al., 2014*), and can promote proliferation and invasion when overexpressed in human cell lines (*Banerjee et al., 2014*). The source of *TRIP13*'s oncogenic activity is unknown, but may stem from effects on chromosome structure and DNA repair pathways (as its meiotic functions would suggest), or may instead arise from aberrant regulation of the SAC.

Pch2/TRIP13 is thus directly implicated in the regulation of HORMA domain-mediated signaling in two separate pathways, meiotic recombination and the SAC. The mechanistic basis for this regulation, however, remains unknown. Here, we show that Pch2/TRIP13 comprises a new family of AAA+ ATPase protein remodelers, with a substrate-recognition domain similar to the NSF/p97/PEX1 remodeler family and a physical mechanism closely related to the bacterial ClpX unfoldase. We show that TRIP13 converts closed, active MAD2 to its inactive open conformer, and that p31(comet) functions as an adapter to recognize closed MAD2 and deliver it to TRIP13. Thus, TRIP13 regulates the SAC through MAD2 conformational conversion and safety belt disengagement, and a similar mechanism for HORMAD complex disassembly likely underlies the enzyme's regulatory functions in meiosis.

## Results

### Structure of *Caenorhabditis elegans* PCH-2

Pch2/TRIP13 proteins are members of the functionally diverse AAA+ ATPase family (*Erzberger and Berger, 2006*; *Wendler et al., 2012*). These proteins share a common architecture, with a family-specific N-terminal domain (NTD) responsible for localization or substrate recognition, and one or two AAA+ ATPase modules that typically assemble into a hexameric ring. AAA+ ATPases are extremely diverse and include DNA and RNA helicases, DNA replication initiators, and a large family termed the 'classic remodelers,' which disaggregate or unfold proteins; these include the SNARE complex disassembly factor NSF, the ubiquitin-directed disaggregase p97/Cdc48, and the ATPase component of the eukaryotic proteasome (*Erzberger and Berger, 2006*).

Sequence comparisons of the Pch2/TRIP13 AAA+ ATPase module fail to clearly classify it within any well-characterized AAA+ family (*Figure 1A*). Moreover, sequence comparisons of the Pch2/TRIP13 NTD fail to identify homology to any known proteins. Therefore, we took a structural approach to determine the relationship of Pch2/TRIP13 to other AAA+ ATPases. We overexpressed and purified *Mus musculus* TRIP13 and its *C. elegans* ortholog PCH-2, and found that while TRIP13 adopts a range of oligomeric states from monomer to hexamer, PCH-2 forms a stable hexamer both with and without added nucleotides (*Figure 1C,D*). We next performed negative-stain electron microscopy (EM) on PCH-2; low-resolution class averages reveal a distinctly asymmetric hexamer in the absence of nucleotides, which becomes more symmetric and compact when ATP is added (*Figure 1E*, *Figure 1—figure supplement 1*). We attempted crystallization both in the presence and absence of nucleotides, and successfully determined the crystal structure of PCH-2 without added nucleotide to a resolution of 2.3 Å. The structure reveals an elongated hexamer with an approximate 'dimer of trimers' symmetry and an overall shape similar to our EM class averages of this state (*Figure 2A*, *Table 1*).

While the PCH-2 NTD (residues 1–99 of 424) lacks detectable sequence homology to other proteins, the structure of this domain shows a clear relationship to the N-terminal substrate recognition domains of a AAA+ 'classic remodeler' sub-family that includes NSF, p97, and PEX1. These proteins possess two-part NTDs with tightly associated N-N and N-C subdomains (*May et al., 1999*; *Yu et al., 1999*). A hydrophobic cleft between the two subdomains binds either directly to substrates, or alternatively to 'adapter' proteins that aid localization and specific substrate recognition (*Kloppsteck et al., 2012*). In PCH-2, the NTD contains a single folded domain similar to the NSF/p97/PEX1 N-C subdomain (*Figure 2B*), and as such does not share these proteins' substrate-binding hydrophobic cleft. Nonetheless, the similarity in NTD structure indicates a hitherto unappreciated evolutionary link between Pch2/TRIP13 and the NSF/p97/PEX1 remodeler family (*Figure 2C*), and strongly suggests that the PCH-2 NTD is involved in substrate recognition, either directly or indirectly through one or more adapter proteins.

PCH-2's single AAA+ ATPase module is composed of two domains, termed the large and small AAA domains. Structural comparisons using the DALI server (*Holm and Rosenström, 2010*) indicate

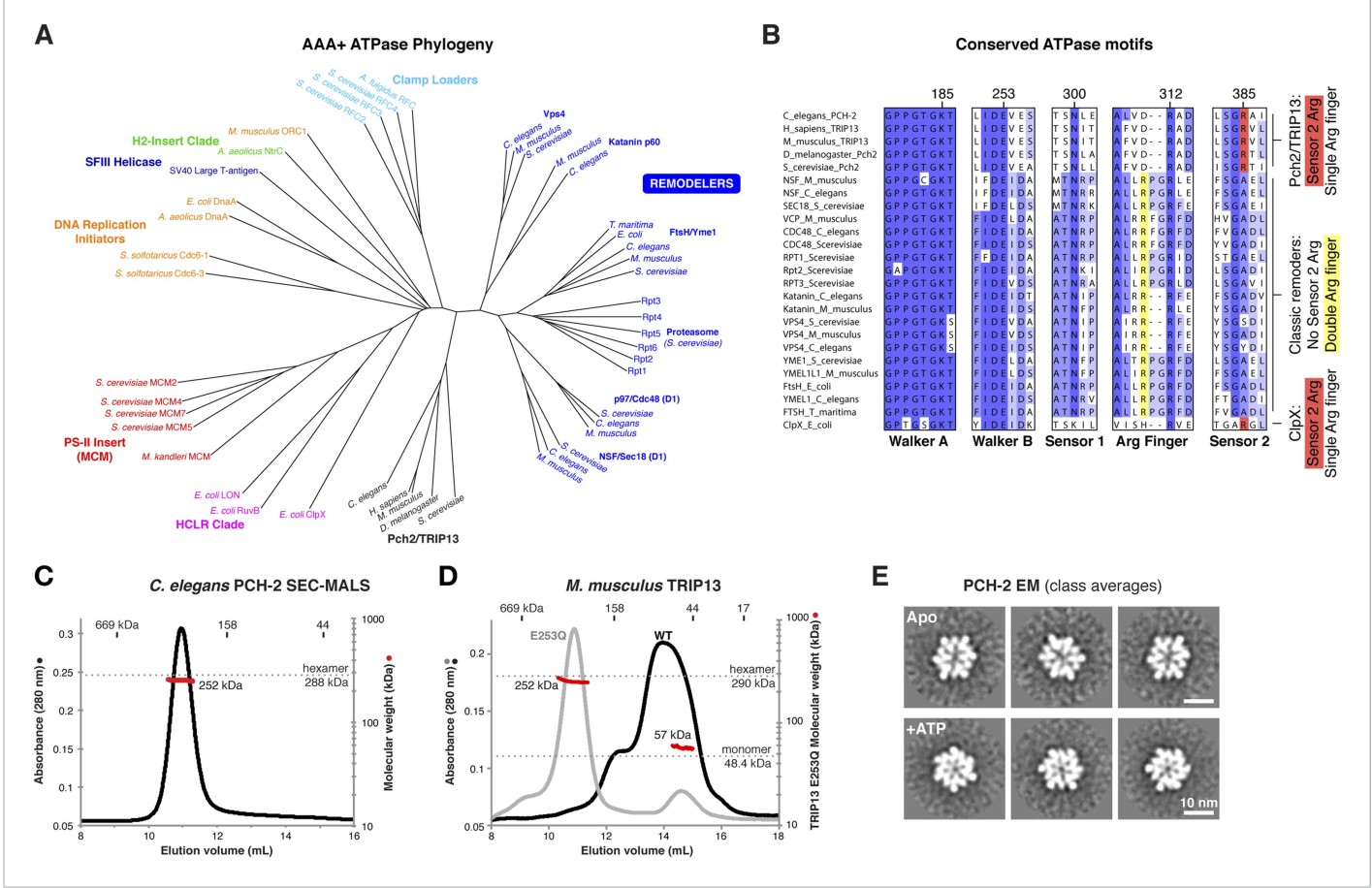

**Figure 1**. PCH-2/TRIP13 is a distinct class of hexmeric AAA+ ATPase. (**A**) Phylogenetic tree of selected AAA+ ATPases, colored by clade (*Erzberger and Berger, 2006*). (**B**) Conserved AAA+ sequence motifs in Pch2/TRIP13, the 'classic remodelers', and *E. coli* ClpX. Pch2/TRIP13 and ClpX lack the first of two conserved arginine residues in the Arg finger region (yellow), and possess a Sensor 2 arginine (R385, red), which the classic remodelers lack. (**C**) Size-exclusion chromatography coupled to multi-angle light scattering (SEC-MALS) analysis of *C. elegans* PCH-2 in the absence of nucleotides. Hexamer molecular weight = 288.1 kDa; measured molecular weight = 252 kDa (red line). (**D**) SEC-MALS analysis of *M. musculus* TRIP13 in the absence of nucleotides. The wild-type protein (black) adopts a mixture of oligomeric states from monomer to hexamer, consistent with findings from *S. cerevisiae* Pch2 (*Chen et al., 2014*). The proportion of higher-molecular weight oligomers increases upon the addition of ATP or non-hydrolyzable analogs (not shown). The ATP hydrolysis-defective TRIP13^E253Q mutant (gray, molecular weight measurements red) is predominantly hexameric both in the presence (shown) and absence of ATP. Molecular weight measurements by SEC-MALS (red) are shown for TRIP13^E253Q; WT measurements (not shown) are consistent. (**E**) Selected negative-stain EM class averages of *C. elegans* PCH-2 without added nucleotides (Apo) or with added ATP. For example raw images, see *Figure 1—figure supplement 1*.

The following figure supplement is available for figure 1:

**Figure supplement 1**. Negative-stain electron microscopy (EM) of *C. elegans* PCH-2.

that the ATP-binding large AAA domain of PCH-2 (residues 100–323) is most structurally similar to the 'classic remodelers,' including NSF/p97/PEX1, Vps4, Katanin p60, and the proteasome ATPase subunits (2.7–3.4 Å r.m.s.d. comparing 150–170 Cα atoms). The domain also shows strong similarity to other AAA+ ATPase families including the 'HCLR' clade that includes ClpX, the unfoldase component of the bacterial ClpXP protease (3.2–3.3 Å r.m.s.d. comparing ~150 Cα atoms) (*Glynn et al., 2009*; *Sauer and Baker, 2011*). The small AAA domain (residues 324–424) is most similar to 'classic remodeler' family members (1.5–2.0 Å r.m.s.d. comparing 60–70 Cα atoms).

As in other AAA+ ATPases, the PCH-2 hexamer assembles through interactions between each subunit's large AAA domain and the small AAA domain of a neighboring subunit, with the ATP-binding sites situated near the subunit interfaces (*Figure 2A,D*). Although no nucleotides were added

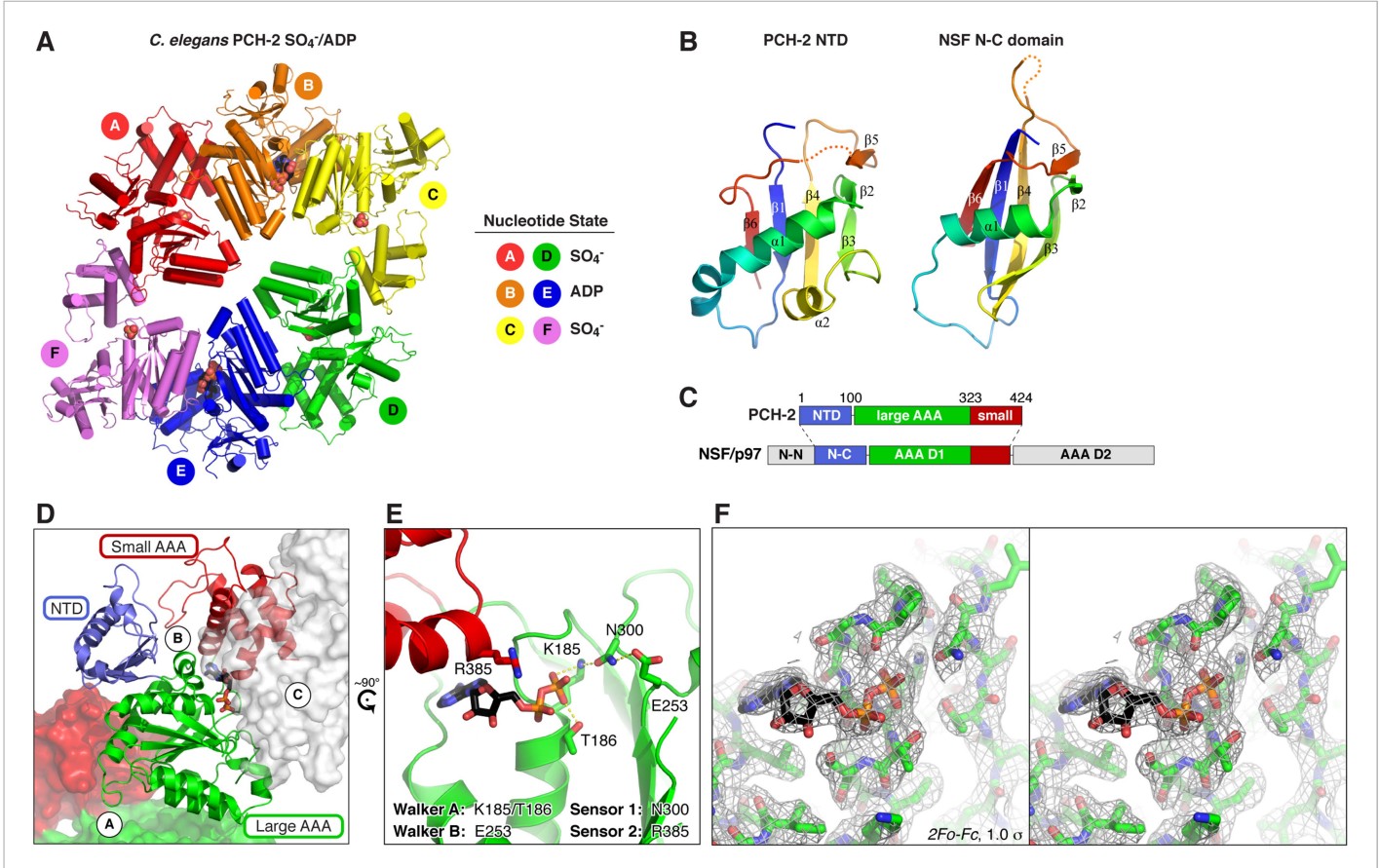

**Figure 2**. Structure of *C. elegans* PCH-2. (**A**) Overall structure of PCH-2. The hexamer shows a 'dimer of trimers' symmetry with chains A/B/C equivalent to chains D/E/F. Chains A/D and C/F are bound to $SO_4^-$ in the ATPase active site, and chains B/E are bound to ADP (space-fill representation). For data collection and refinement statistics, see *Table 1*. (**B**) Structural comparison of the PCH-2 NTD (residues 1–99) with the NSF N-C subdomain (residues 92–189; [*May et al., 1999*]); Cα r.m.s.d. 1.94 Å over 55 residues. (**C**) Schematic of Pch2/TRIP13 domain structure vs NSF/p97. Pch2/TRIP13 shares these proteins' N-C subdomain (blue) and one of their tandem AAA+ ATPase regions (green/red). (**D**) Close-up view of PCH-2 chain B, with domains colored as in (**C**), showing its packing against subunits A and C. (**E**) Close-up of ADP bound to PCH-2 chain B, with AAA+ motifs shown as sticks. For close-up views of all six active sites, see *Figure 2—figure supplement 1*. (**F**) Stereo view of refined $2F_o − F_c$ electron density at 2.3 Å resolution, contoured at 1.0 σ, for the bound ADP and surrounding residues in PCH-2 chain B. View is equivalent to (**E**); the small AAA domain has been removed for clarity.

The following figure supplement is available for figure 2:

**Figure supplement 1**. X-ray crystallographic analysis of *C. elegans* PCH-2.

during purification or crystallization of PCH-2, we observed that two subunits in the hexamer (chains B and E) are bound to ADP (*Figure 2A,E*, *Figure 2—figure supplement 1B*), enabling a close analysis of PCH-2 active site structure. PCH-2 possesses the characteristic Walker A, Walker B, Sensor-1, and 'arginine finger' motifs in the large AAA domain that cooperate to bind nucleotide (*Figures 1B, 2E*) (*Wendler et al., 2012*). In many AAA+ ATPases, nucleotide binding is sensed by an additional 'Sensor-2' motif, typically an arginine residue, reaching from the small AAA domain into the active site. This motif is involved in nucleotide binding, hydrolysis, and nucleotide-dependent inter-domain conformational changes in various AAA+ ATPases (*Ogura et al., 2004*). Curiously, the 'classic remodelers' family, including NSF/p97/PEX1, uniformly lacks the Sensor-2 motif and also possesses a second arginine adjacent to the arginine finger (*Figure 1B*). These differences indicate that this family's mechanism for ATP-driven conformational changes may have diverged somewhat from other AAA+ ATPases (*Ogura et al., 2004*; *Erzberger and Berger, 2006*). Pch2/TRIP13 proteins, in contrast, possesses only a single arginine finger (R312 in *C. elegans* PCH-2), and our PCH-2 structure shows that a conserved arginine (R385) is properly positioned to act as a Sensor-2 motif (*Figures 1B, 2E*). This

**Table 1**. Data collection and refinement statistics

| | PCH-2 SeMet | PCH-2 native |
|---|---|---|
| Data collection | | |
| Synchrotron/Beamline | APS 24ID-E | SSRL 12-2 |
| Resolution (Å) | 3.23 | 2.3 |
| Wavelength (Å) | 0.97921 | 0.9795 |
| Space group | C222$_1$ | C222$_1$ |
| Unit cell dimensions (a, b, c) Å | 126.1, 239.5, 198.2 | 126.7 241.0 197.9 |
| Unit cell angles (α, β, γ) ° | 90, 90, 90 | 90, 90, 90 |
| $I/\sigma$ (last shell) | 9.3 (1.0) | 17.9 (0.8) |
| * $R_{sym}$ (last shell) | 0.198 (2.166) | 0.098 (3.143) |
| † $R_{meas}$ (last shell) | 0.213 (2.326) | 0.102 (3.297) |
| ‡ Isotropic CC$_{1/2}$, last shell | 0.592 | 0.275 |
| § Directional CC$_{1/2}$, last shell (Å) | | |
| a* | – | 0.498 (2.3 Å) |
| b* | – | 0.532 (2.3 Å) |
| c* | – | 0.608 (3.2 Å) |
| Completeness (last shell) % | 99.9 (99.9) | 99.5 (90.8) |
| Number of reflections | 33,462 | 1,808,343 |
| *unique* | 4410 | 134,133 |
| Multiplicity (last shell) | 7.5 (7.6) | 13.5 (10.6) |
| Number of sites | 68 | – |
| § Anisotropic scaling B-factors (Å$^2$) | | |
| a* | – | −8.09 |
| b* | – | −8.02 |
| c* | – | 16.11 |
| isotropic B-factor correction | – | −19.65 |
| Refinement | | |
| Resolution range (Å) | – | 40 - 2.3 |
| No. of reflections | – | 96,084 |
| *working* | – | 91,200 |
| *free* | – | 4884 |
| # $R_{work}$ (%) | – | 22.97 |
| # $R_{free}$ (%) | – | 26.42 |
| Structure/Stereochemistry | | |
| Number of atoms | – | 18,017 |
| *ligands (ADP, SO$_4$)* | – | 89 |
| *solvent* | – | 55 |
| r.m.s.d. bond lengths (Å) | – | 0.004 |
| r.m.s.d. bond angles (°) | – | 0.730 |
| ¶ PDB ID | – | 4XGU |

*$R_{sym} = \sum\sum_j |I_j - \langle I \rangle| / \sum I_j$, where $I_j$ is the intensity measurement for reflection j and $\langle I \rangle$ is the mean intensity for multiply recorded reflections.

†$R_{meas} = \sum_h [\sqrt{(n/(n-1))} \sum_j [I_{hj} - \langle I_h \rangle]/\sum_{hj} \langle I_h \rangle$ where $I_{hj}$ is a single intensity measurement for reflection h, $\langle I_h \rangle$ is the average intensity measurement for multiply recorded reflections, and $n$ is the number of observations of reflection h.

‡CC$_{1/2}$ is the Pearson correlation coefficient between the average measured intensities of two randomly-assigned half-sets of the measurements of each unique reflection (**Karplus and Diederichs, 2012**).

§High-resolution native data were anisotropically scaled and elliptical data cutoffs were applied according to directional intensity and $CC_{1/2}$ data (see 'Materials and methods' and **Figure 3—figure supplement 1A** for details on data anisotropy and resolution cutoffs).

#$R_{work, free} = \sum ||F_{obs}| - |F_{calc}||/|F_{obs}|$, where the working and free R-factors are calculated using the working and free reflection sets, respectively.

¶Coordinates and structure factors have been deposited in the RCSB Protein Data Bank (www.pdb.org).

finding suggests that despite sharing a common NTD with a family of 'classic remodelers', the detailed mechanism for nucleotide-dependent conformational changes in Pch2/TRIP13 may more closely resemble other AAA+ families.

## Nucleotide-driven conformational changes in the PCH-2 hexamer

The distinctly asymmetric hexamer architecture of PCH-2 provides clues to conformational changes that likely occur during ATP binding, hydrolysis, and release. Within the hexamer, four PCH-2 subunits adopt a 'closed' conformation equivalent to that observed in most AAA+ 'classic remodeler' structures, with the large and small AAA domains tightly associated around the ATP-binding site. Two of these subunits (chains B and E) are bound to ADP in our structure, while the other two (chains A and D) contain a $SO_4^-$ ion from the crystallization buffer. Asymmetry in the PCH-2 hexamer arises from large conformational differences in the remaining two subunits (chains C and F, also bound to $SO_4^-$), situated on opposite ends of the extended hexamer. Compared to the four closed subunits, these chains adopt an 'open' conformation, in which the small AAA domain is rotated ~70° away from the large AAA domain (**Figure 3A–B**).

The dramatic conformational differences between subunits, and the resulting overall hexamer architecture of PCH-2, are distinct from most existing structures of AAA+ 'classic remodelers', which are typically either symmetric or display a subtle helical pitch resulting in a 'lock-washer' conformation, and usually lack the significant rotations between large and small AAA domains seen in PCH-2 (**Davies et al., 2008**; **Lander et al., 2012**; **Zhao et al., 2015**). Instead, the PCH-2 structure closely resembles several prior structures of the ClpX unfoldase, which contain four 'closed' and two 'open' subunits arranged in the same pattern as in PCH-2 (**Figure 3C,F,G**) (**Glynn et al., 2009**; **Stinson et al., 2013**). This conformation was observed both in the absence of nucleotides (**Glynn et al., 2009**) and in the presence of a non-hydrolyzable ATP analog (ATP-γ-S), which was found to bind the four 'closed' subunits but not the two 'open' subunits (**Stinson et al., 2013**). Detailed biochemical analysis of the ClpX mechanism has demonstrated that sequential ATP binding, hydrolysis, and release drive cyclical open → closed → open conformational changes within each subunit (**Glynn et al., 2009**; **Stinson et al., 2013**). These motions in turn drive axial movement of loops lining the hexamer pore (pore loops), which contain aromatic residues that directly engage substrate proteins during unfolding (**Siddiqui et al., 2004**; **Iosefson et al., 2015**). Despite a potentially diverged mechanism for ATP-powered conformational changes, the classic remodelers possess functionally equivalent pore loops (**Zhao et al., 2010**). Within each half-hexamer of PCH-2 (chains A/B/C and D/E/F), the pore loops (residues 217–226; **Figure 3H**) are axially staggered to create a 'spiral staircase' of likely substrate-engaging groups. We interpret these pore loop positions as representing structural intermediates adopted during ATP binding, hydrolysis, and release within each PCH-2 subunit that drive substrate remodeling (**Figure 3I**).

## ATP binding and hydrolysis in PCH-2 and *M. musculus* TRIP13

To test the physical mechanism of Pch2/TRIP13 and the roles of active-site and pore-loop residues, we measured nucleotide binding and hydrolysis by PCH-2 and its *M. musculus* ortholog TRIP13. As in other AAA+ ATPases, a mutation in the Walker B motif of both PCH-2 and TRIP13 (E253Q in both enzymes) retains high-affinity nucleotide binding, while PCH-2 Walker A and Sensor-1 mutants do not bind nucleotide (**Figure 4A,B**). We found that TRIP13$^{E253Q}$ also forms stable hexamers, in contrast to the predominantly monomeric wild-type enzyme (**Figure 1D**). Mutation of PCH-2 R385 also results in the loss of nucleotide binding, illustrating that this residue is likely to be functionally analogous to the Sensor-2 motifs in other AAA+ ATPases.

We used an enzyme-coupled assay to measure ATP hydrolysis by PCH-2 and TRIP13. Both enzymes showed modest but reproducible ATPase activity, and mutation of conserved active site residues

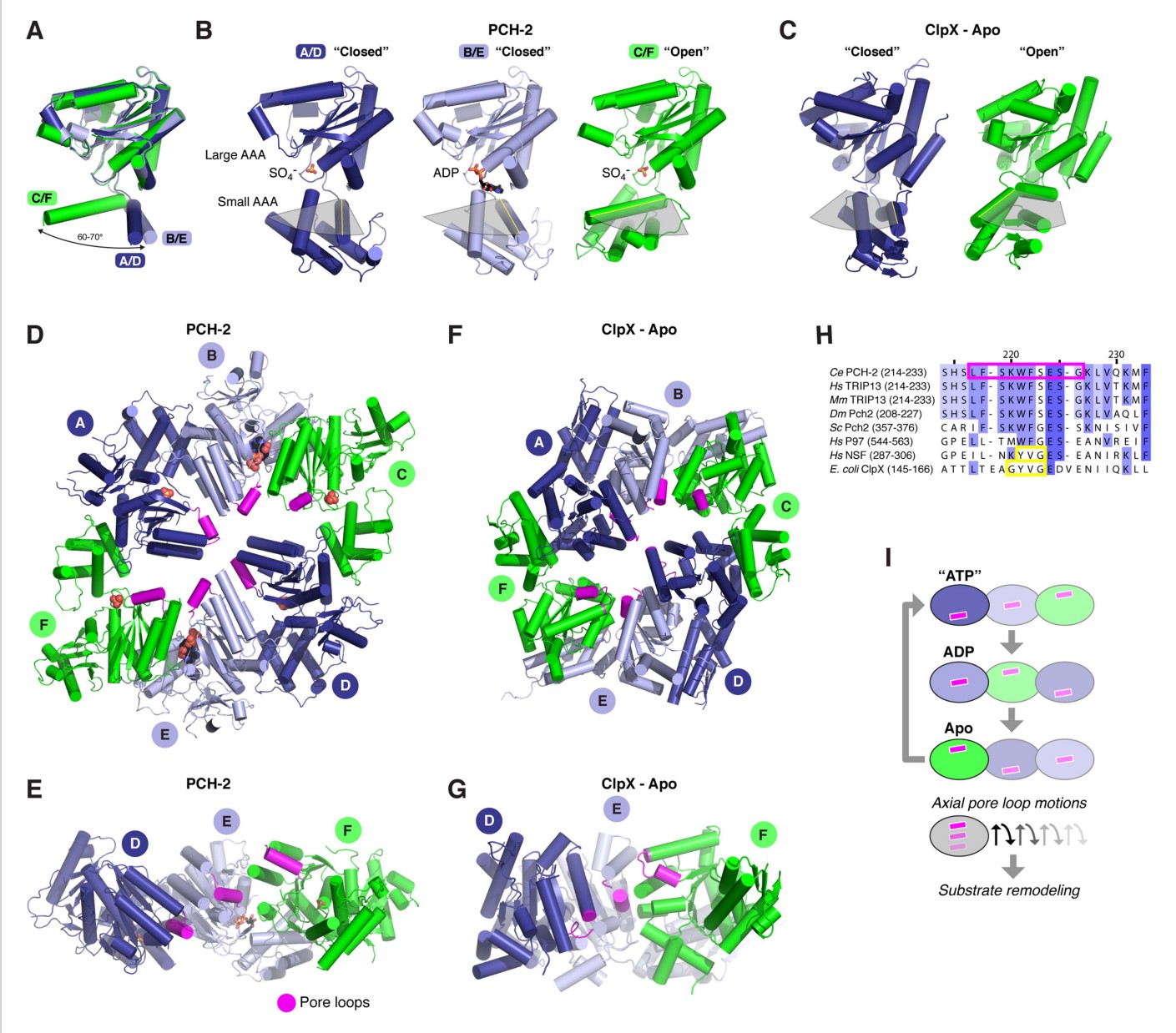

Figure 3. Conformational changes within the PCH-2 hexamer. (A) Structural basis for nucleotide binding-dependent conformational changes. All six subunits (A/D dark blue, B/E light blue, C/F green) are overlaid based on their large AAA domains, and their small AAA domains are represented by a single α-helix, residues 323–342. (B) Relative orientation of large and small AAA domains in different subunit types. Bound ADP and SO4− ions are shown as sticks. While the small AAA domain position varies widely between subunits, all six subunit–subunit interfaces are equivalent, forming six rigid-body units within the hexamer (see *Figure 3—figure supplement 1*). (C) 'Closed' (blue) and 'open' (green) ClpX monomers in the nucleotide-free ClpX hexamer (PDB ID 3HTE; [*Glynn et al., 2009*]). Later work showed that the 'closed' conformation is compatible with nucleotide binding (*Stinson et al., 2013*). (D) Top view of the asymmetric PCH-2 hexamer, with subunits colored as in (A) and (B), and pore loops (residues 217–226) colored magenta. (E) Pore-side view of PCH-2 D/E/F chains (A/B/C chains removed), showing the axial staggering of these subunits' pore loops. (F) Top view of the nucleotide-free ClpX hexamer (*Glynn et al., 2009*), with 'closed' and 'open' subunits colored as in PCH-2 and pore loops (residues 145–153) colored magenta. (G) Pore-side view of ClpX D/E/F chains (A/B/C removed), colored as in (F). (H) Sequence alignment of pore loop region in PCH-2 orthologs, and equivalent region of human p97 and NSF, and *E. coli* ClpX. Magenta box: PCH-2 pore loop; Yellow boxes: NSF 'YVG' and ClpX 'GYVG' motifs. (I) Schematic model for ATP-driven conformational changes in PCH-2, with pore-side view equivalent to panel F. As the left-most subunit binds ATP (blue; represented by the closed 'ATP'-like state in chain D), hydrolyzes ATP to ADP (light blue; represented by

*Figure 3. continued on next page*

*Figure 3. Continued*

PCH-2 chain E), then releases hydrolyzed ADP (green; represented by PCH-2 chain F), its pore loop (magenta) undergoes axial motions that drive substrate remodeling.
The following figure supplement is available for figure 3:

**Figure supplement 1**. The asymmetric PCH-2 hexamer is composed of equivalent rigid-body groups.

resulted in the complete loss of activity (*Figure 4C–E*). We next tested the importance of the enzymes' pore loops, which in other AAA+ ATPases are necessary to properly couple ATP hydrolysis to substrate engagement (*Siddiqui et al., 2004*). We generated alanine mutants of PCH-2 W221 and F222, which align with the ClpX/NSF 'YVG'/'GYVG' motifs (*Figure 3H*), and also created a 'pore loop AG' mutant (PCH-2$^{AG}$) in which residues 218-227 were replaced by an Ala-Gly linker of equal length. These mutants all formed soluble hexamers as in wild-type PCH-2, but showed variable ATPase activity: while PCH-2$^{F222A}$ showed a fivefold reduction in basal ATPase activity, PCH-2$^{W221A}$ and PCH-2$^{AG}$ showed a striking increase in activity, with the $k_{cat}$ of PCH-2$^{W221A}$ around threefold higher than that of the wild-type enzyme (*Figure 4C,E*). Similarly, *M. musculus* TRIP13$^{W221A}$ had a more than twofold higher basal $k_{cat}$ than wild-type TRIP13 (*Figure 4D,E*). The variable effect of pore loop mutations in PCH-2/TRIP13 suggests that this element may play a role in coupling ATP hydrolysis to substrate engagement.

## p31(comet) functions as an adapter between MAD2 and TRIP13

Timely SAC inactivation relies on both TRIP13 and p31(comet), and prior work has shown that the two proteins can together dissociate MAD2:CDC20 complexes in vitro (*Eytan et al., 2014*). As p31(comet) is known to bind both MAD2 (*Habu et al., 2002*; *Yang et al., 2007*) and TRIP13 (*Tipton et al., 2012*), we reasoned that the protein may act as an adapter. We mapped sequence conservation onto the structure of human p31(comet):MAD2 (*Yang et al., 2007*), and identified a highly conserved surface on p31(comet) opposite its MAD2-binding interface (*Figure 5A*). To test whether this surface, which includes residues on the p31(comet) 'safety-belt' and a short loop bordering this motif, is responsible for TRIP13 interaction, we generated a series of mutations in *M. musculus* p31(comet). Several mutations to this conserved surface disrupted TRIP13 binding in a yeast two-hybrid assay, while retaining MAD2 binding (*Figure 5B*). Conversely, a previously-characterized mutant at the MAD2 interface (*Yang et al., 2007*) disrupted MAD2 binding but did not affect the interaction with TRIP13 (*Figure 5B,D*). To test whether p31(comet) is able to simultaneously interact with MAD2 and TRIP13, we performed a yeast three-hybrid assay. This assay showed an interaction between TRIP13 and MAD2 that depends on the presence of untagged p31(comet) (*Figure 5C*), showing that p31(comet) can indeed function as an adapter between MAD2 and TRIP13 (*Figure 5G*).

We next sought to reconstitute the TRIP13:p31(comet):MAD2 complex in vitro. We separately purified *M. musculus* TRIP13$^{E253Q}$, which is catalytically inactive and forms more stable hexamers than wild-type TRIP13 (*Figure 1D*), wild-type p31(comet), and a monomeric variant of MAD2 (MAD2$^{R133A}$) in which open (O-MAD2) and closed (C-MAD2) monomers can be separately purified (*Figure 5—figure supplement 2*) (*Sironi et al., 2001*; *Luo et al., 2004*; *Mapelli et al., 2006*). We mixed TRIP13$^{E253Q}$, p31(comet), and C-MAD2$^{R133A}$ in the presence of ATP, and measured complex formation using size-exclusion chromatography. In agreement with prior studies (*Yang et al., 2007*), p31(comet) and C-MAD2$^{R133A}$ formed a stable heterodimeric complex. When this complex was pre-incubated with TRIP13$^{E253Q}$, a small amount of both proteins was shifted into the TRIP13 hexamer peak (*Figure 5E*, *Figure 5—figure supplement 1*). Semi-quantitative analysis of Coomassie-stained gels, and light-scattering based molecular-weight measurements on this peak, revealed that for each TRIP13$^{E235Q}$ hexamer, about 1 copy of p31(comet):C-MAD2$^{R133A}$ was shifted into the TRIP13 peak (*Figure 5E,F*, *Figure 5—figure supplement 1*). MAD2 did not shift in the absence of p31(comet), and only a very small amount of p31(comet) shifted in the absence of MAD2 (*Figure 5—figure supplement 1*). p31(comet) mutants that disrupt either MAD2 or TRIP13 binding also largely eliminated p31(comet):MAD2 co-migration with TRIP13$^{E253Q}$ (*Figure 5—figure supplement 1*). Taken together with our yeast two-hybrid results, these data suggest that a single p31(comet):MAD2

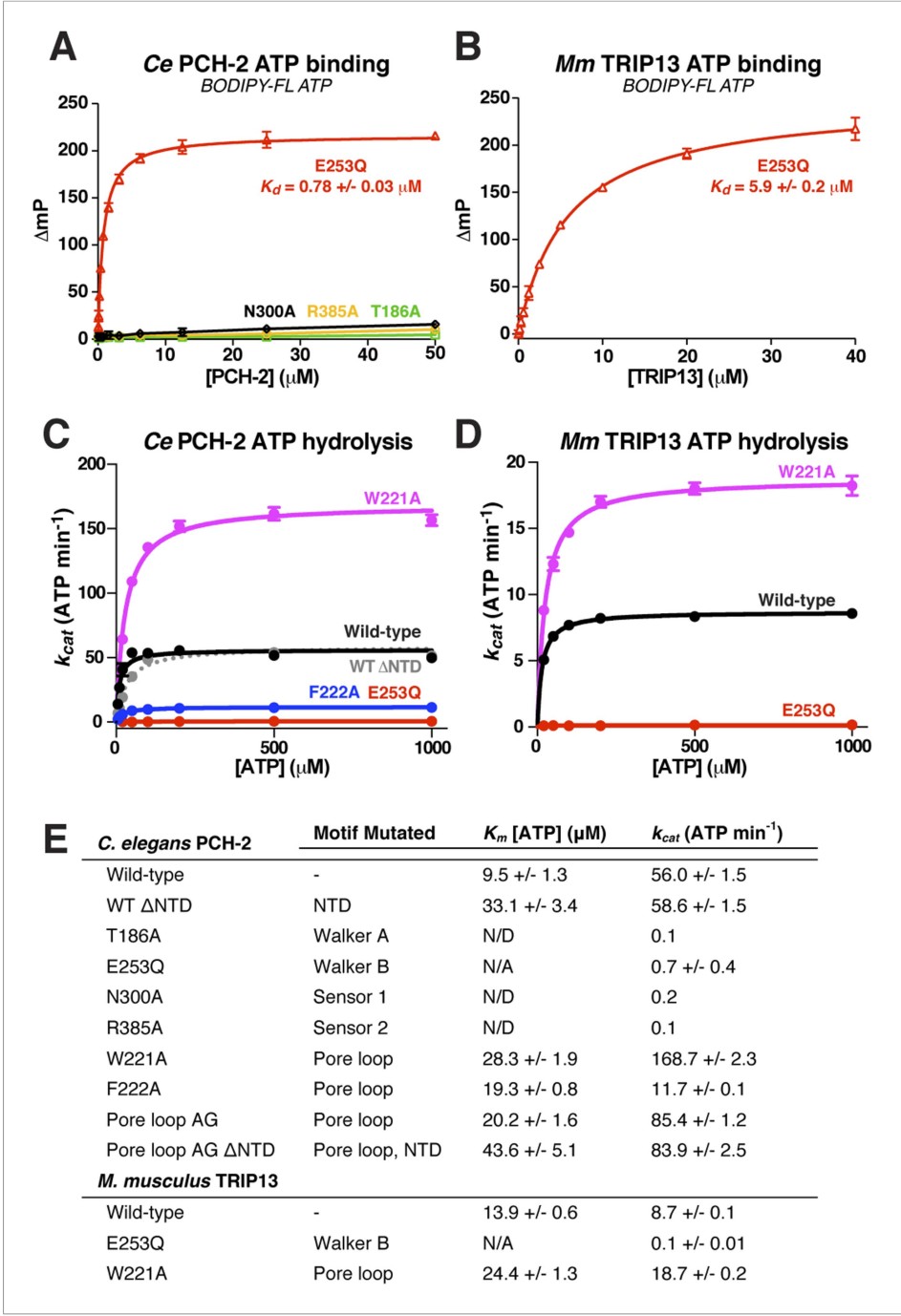

Figure 4. Nucleotide binding and hydrolysis by PCH-2 and TRIP13. (**A**) Binding of PCH-2 active-site mutants to BODIPY-FL ATP. (**B**) Binding of *M. musculus* TRIP13$^{E253Q}$ to BODIPY-FL ATP. (**C**) Basal ATP hydrolysis rates of wild-type and mutant *C. elegans* PCH-2 at pH 8.5 (optimal for ATPase activity; ATPase stimulation assays (*Figure 6*) were performed at pH 7.5, where basal activity is lower but stimulation is more robust). E253Q: Walker B ATPase mutant; W221A/F222A: pore loop mutants; WT ΔNTD: residues 100–424. (**D**) Basal ATP hydrolysis rates of wild-type and mutant *M. musculus* TRIP13 at pH 8.5. Residue numbering for mutants is identical to *C. elegans* PCH-2. (**E**) $K_m/k_{cat}$ values (reported as ATP min$^{-1}$ per hexameric enzyme) for wild-type and mutant PCH-2 and TRIP13. For PCH-2$^{T186A}$, PCH-2$^{N300A}$, and PCH-2$^{R385A}$, rates were measured at a single ATP concentration of 2 mM, so $K_m$ was not determined (N/D). For PCH-2$^{E253Q}$ and TRIP13$^{E253Q}$, very low ATPase activity precluded a reliable $K_m$ determination (N/A).

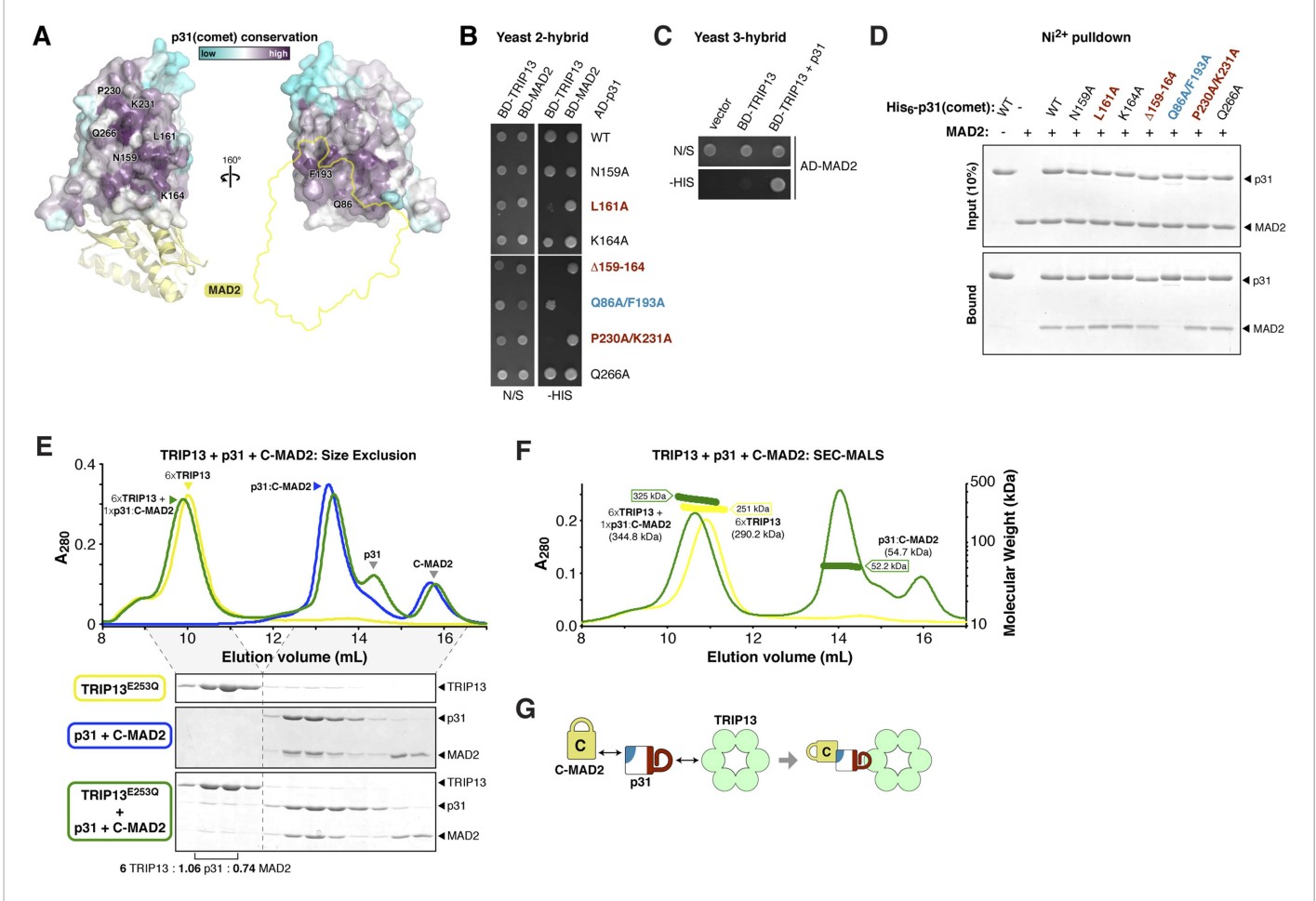

**Figure 5**. p31(comet) functions as an adapter between TRIP13 and MAD2. (**A**) Two views of the crystal structure of human p31(comet) (colored by conservation) bound to MAD2 (yellow) (*Yang et al., 2007*). Residue numbers shown are of *M. musculus* p31(comet) (76% identity with *Homo sapiens* p31 (comet); all noted residues are conserved). (**B**) Yeast two-hybrid assay for *M. musculus* p31(comet) binding to TRIP13 and MAD2. The p31(comet)-MAD2 interaction can also be detected using purified proteins (panel **D**). BD: Gal4 DNA-binding domain fusion; AD: Gal4 activation domain fusion. N/S: no selection; -HIS: selection for interaction between BD- and AD-fused proteins. (**C**) Yeast three-hybrid assay showing interaction of BD-TRIP13 and AD-MAD2 in the presence of untagged p31(comet). (**D**) Ni$^{2+}$-pulldown assay using purified His$_6$-tagged *M. musculus* p31(comet) pulling down untagged *M. musculus* MAD2. (**E**) Size exclusion chromatography traces and gels from *M. musculus* TRIP13$^{E253Q}$ (yellow), p31(comet):C-MAD2$^{R133A}$ (blue), and an equimolar mixture of TRIP13$^{E253Q}$ + p31(comet):C-MAD2$^{R133A}$ (green). Quantitation of Coomassie-stained bands in bottom gel (lanes 2 and 3) show an ~6:1 molar ratio of TRIP13$^{E253Q}$ to p31(comet):MAD2 (6 copies of TRIP13, 1.06 copies of p31(comet) and 0.74 copies of C-MAD2$^{R133A}$). See *Figure 5—figure supplement 1* for full gels and analysis of different protein combinations including p31(comet) mutants, and *Figure 5—figure supplement 2* for purification of p31(comet) and MAD2. (**F**) SEC-MALS analysis of TRIP13$^{E253Q}$ (yellow) and TRIP13$^{E253Q}$:p31(comet):C-MAD2$^{R133A}$ (green). TRIP13$^{E253Q}$ migrates as a single peak with measured molecular weight of 251 kDa, close to the calculated hexamer molecular weight of 290.2 kDa. Upon the addition of p31(comet) and C-MAD2$^{R133A}$, the measured molecular weight shifts to 325 kDa. The shift of 74 kDa is close to the weight of a p31(comet):C-MAD2$^{R133A}$ complex (54.7 kDa). Excess p31(comet):C-MAD2$^{R133A}$ elutes after the complex with TRIP13. (**G**) Schematic illustrating p31(comet) functioning as an adapter between C-MAD2 (via blue surface) and TRIP13 (via red surface). See *Figure 8—figure supplement 1C* for the p31(comet) crystal structure colored equivalently.

The following figure supplements are available for figure 5:

**Figure supplement 1**. Interactions between *M. musculus* TRIP13, p31(comet), and MAD2.

**Figure supplement 2**. Purification and characterization of *M. musculus* MAD2 and p31(comet).

complex associates with a TRIP13 hexamer, first through transient TRIP13-p31(comet) binding (likely mediated by the TRIP13 NTD), then through direct interactions between TRIP13 and MAD2 (at the TRIP13 hexamer pore). We were unable to directly test the role of the TRIP13 pore loops in binding, however, as the TRIP13$^{W221A/E253Q}$ double mutant did not form stable hexamers (data not shown).

## PCH-2/TRIP13 ATPase activity is stimulated by p31(comet) + MAD2

We next examined whether PCH-2 or TRIP13 ATPase activity is stimulated by MAD2, p31(comet), or the p31(comet):MAD2 complex. While PCH-2 ATPase activity was mostly unaffected by the addition of either MAD-2 or the recently-identified p31(comet) ortholog CMT-1 (*Vleugel et al., 2012*), it was modestly stimulated in the presence of both proteins (*Figure 6A*). A PCH-2 construct missing the NTD was not stimulated by the addition of MAD-2 + CMT-1, supporting the proposed role for this domain in substrate recognition (*Figure 6A*). The PCH-2$^{W221A}$ pore-loop mutant was also not stimulated by MAD-2 + CMT-1, supporting the idea that this mutant uncouples ATP hydrolysis from productive substrate engagement (*Figure 6A*).

*M. musculus* TRIP13 was strongly stimulated by the addition of MAD2 + p31(comet) (*Figure 6B*). Stimulation did not depend on the presence of CDC20, but was highly sensitive to MAD2 conformation: while C-MAD2$^{R133A}$ stimulated TRIP13 equivalently to wild-type MAD2, O-MAD2$^{R133A}$ showed minimal stimulation, and the locked-open 'loopless' MAD2 mutant (*Mapelli et al., 2007*) showed no stimulation (*Figure 6B*). As with PCH-2, neither MAD2 nor p31(comet) alone had a stimulatory effect on TRIP13. Also consistent with PCH-2, the TRIP13$^{W221A}$ pore-loop mutant showed a very high basal level of ATP hydrolysis, and was not further stimulated by MAD2 + p31(comet) (*Figure 6B*). Finally, we found that p31(comet) mutations that disrupt either MAD2 or TRIP13 binding also significantly reduce stimulation of TRIP13 by p31(comet) + MAD2 (*Figure 6C,D*).

## TRIP13 catalyzes the conversion of closed MAD2 to the open conformer

The above results suggest a model in which p31(comet) recognizes C-MAD2 and delivers it to TRIP13 for remodeling. If TRIP13 functions by unfolding MAD2, or alternatively converting C-MAD2 to O-MAD2, a prolonged incubation with TRIP13 should eliminate the stimulatory effect of p31(comet) + MAD2. This is indeed the case: we found that the ability of p31(comet) + MAD2 to stimulate TRIP13 ATPase activity was almost eliminated after a 2-hr pre-incubation, in an ATP-dependent manner (*Figure 7A*, compare samples #2, 3, and 4). Importantly, p31(comet) and MAD2 were in sixfold molar excess to TRIP13 hexamers in this assay, illustrating that each TRIP13 hexamer acted on multiple p31 (comet):MAD2 complexes during the pre-incubation period. Addition of fresh MAD2 to the pre-incubated samples rescued TRIP13 stimulation (*Figure 7A*, sample #5). We interpret these findings to indicate that TRIP13 is stimulated by the p31(comet):MAD2 complex but acts on MAD2 specifically, likely converting C-MAD2 into the non-stimulatory open state (*Figure 7B*).

To directly assay whether TRIP13 converts C-MAD2 to O-MAD2, we took advantage of the fact that in the dimerization-defective MAD2$^{R133A}$ mutant, the open and closed conformers are separable by anion-exchange chromatography (*Figure 7C*, *Figure 5—figure supplement 2A*) (*Luo et al., 2004*). When we incubated p31(comet) + C-MAD2$^{R133A}$ with TRIP13, the p31(comet):C-MAD2 complex that initially forms was dissociated, and C-MAD2$^{R133A}$ was converted to the open conformer (*Figure 7D*). This conversion depended on active TRIP13, ATP, and p31(comet). Despite a high rate of ATP hydrolysis, the TRIP13$^{W221A}$ pore-loop mutant was unable to catalyze MAD2 conversion, illustrating that pore loop integrity is critical for MAD2 conformational conversion. To directly measure the catalytic activity of TRIP13, we titrated the enzyme and monitored C-MAD2$^{R133A}$ to O-MAD2$^{R133A}$ conversion. At 37°C, a single TRIP13 hexamer catalyzes the conversion of 1.9 ± 0.2 MAD2 molecules per minute (*Figure 7E*). Combining this measurement with TRIP13's fully-stimulated ATPase activity in these conditions (16.7 ± 0.3 ATP min$^{-1}$ per hexamer; *Figure 7G*), we estimate that TRIP13 hydrolyzes 8–10 ATP's per MAD2 conformational conversion. This number is similar to several prior measurements of NSF-mediated SNARE complex disassembly (6–50 ATP per event, depending on experimental conditions) (*Cipriano et al., 2013*; *Ryu et al., 2015*; *Shah et al., 2015*) or ClpX-mediated unfolding of a small model substrate (~150 ATP) (*Burton et al., 2001*), likely reflecting that MAD2 conformational conversion may require only a local perturbation of the safety-belt motif, rather than complete unfolding (see 'Discussion'). This idea would fit with recent work indicating that NSF unfolds its substrates in a single step, using the energy from multiple ATP hydrolysis events to build up tension within the hexamer, then promoting a critical conformational change that results in SNARE

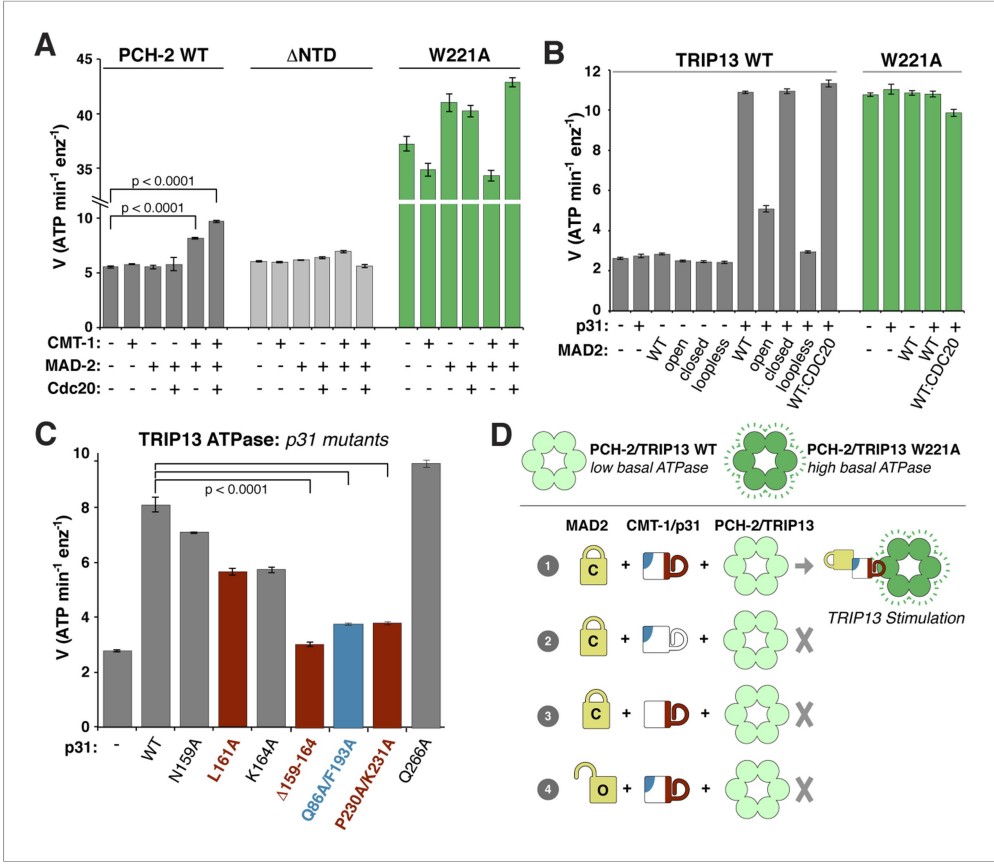

**Figure 6**. ATP hydrolysis in PCH-2/TRIP13 is stimulated by p31(comet) + MAD2. (**A**) Stimulation of *C. elegans* PCH-2 ATPase activity by CMT-1 (p31(comet)) and MAD-2. See *Figure 6—figure supplement 1* for purification of *C. elegans* CMT-1 and MAD-2. Cdc20: N-terminal MBP fusion of *C. elegans* FZY-1 residues 98-140. Substrates were in sixfold molar excess of PCH-2 hexamer. (**B**) Stimulation of *M. musculus* TRIP13 ATPase activity by p31(comet) and MAD2. WT: wild-type MAD2 dimer; 'open': O-MAD2$^{R133A}$ monomer; 'closed': C-MAD2$^{R133A}$ monomer; 'loopless': residues 109–117 replaced by GSG (adopts monomeric open form) (*Mapelli et al., 2007*). CDC20: N-terminal MBP fusion of CDC20 residues 111–150, sufficient for MAD2 binding (*Luo et al., 2000*). Substrates were in sixfold molar excess of TRIP13 hexamer. (**C**) Stimulation of TRIP13 ATPase activity in the presence of p31(comet) mutants. (**D**) Schematic illustrating requirements for TRIP13 stimulation. PCH-2/TRIP13 is stimulated by the combination of C-MAD2 and p31(comet) (scheme 1); mutation of either binding surface of p31(comet) (schemes 2 and 3) eliminates stimulation, as does replacement of C-MAD2 with O-MAD2 (scheme 4). p-values in (**A**) and (**C**) were calculated using an unpaired Student's T-test.

The following figure supplement is available for figure 6:

**Figure supplement 1**. Purification and characterization of *C. elegans* MAD-2 and CMT-1.

---

complex disassembly (*Ryu et al., 2015*). A similar 'spring-loaded' mechanism in TRIP13 could enable the enzyme to catalyze conversion/disassembly of unliganded C-MAD2 or its complexes with partner proteins such as CDC20 with similar efficiency; alternatively, if MAD2 conformational conversion requires several rounds of ATP hydrolysis, the energy requirements for disassembly of ligand-stabilized MAD2 may be significantly higher than the 8–10 ATP we measured for C-MAD2$^{R133A}$ conformational conversion.

Finally, we tested the ability of p31(comet) mutants that disrupt binding to either MAD2 or TRIP13 to support MAD2 conformational conversion. The MAD2-binding mutant (Q86A/F193A) and two TRIP13-binding mutants (Δ159–164 or P230A/K231A) each modestly reduced MAD2 conversion compared to wild-type p31(comet) (*Figure 7F*). Combinations of these mutants, however, almost completely eliminated MAD2 conversion (*Figure 7F*), illustrating that the adapter function of p31(comet) is critical for TRIP13 to recognize and convert C-MAD2 to O-MAD2.

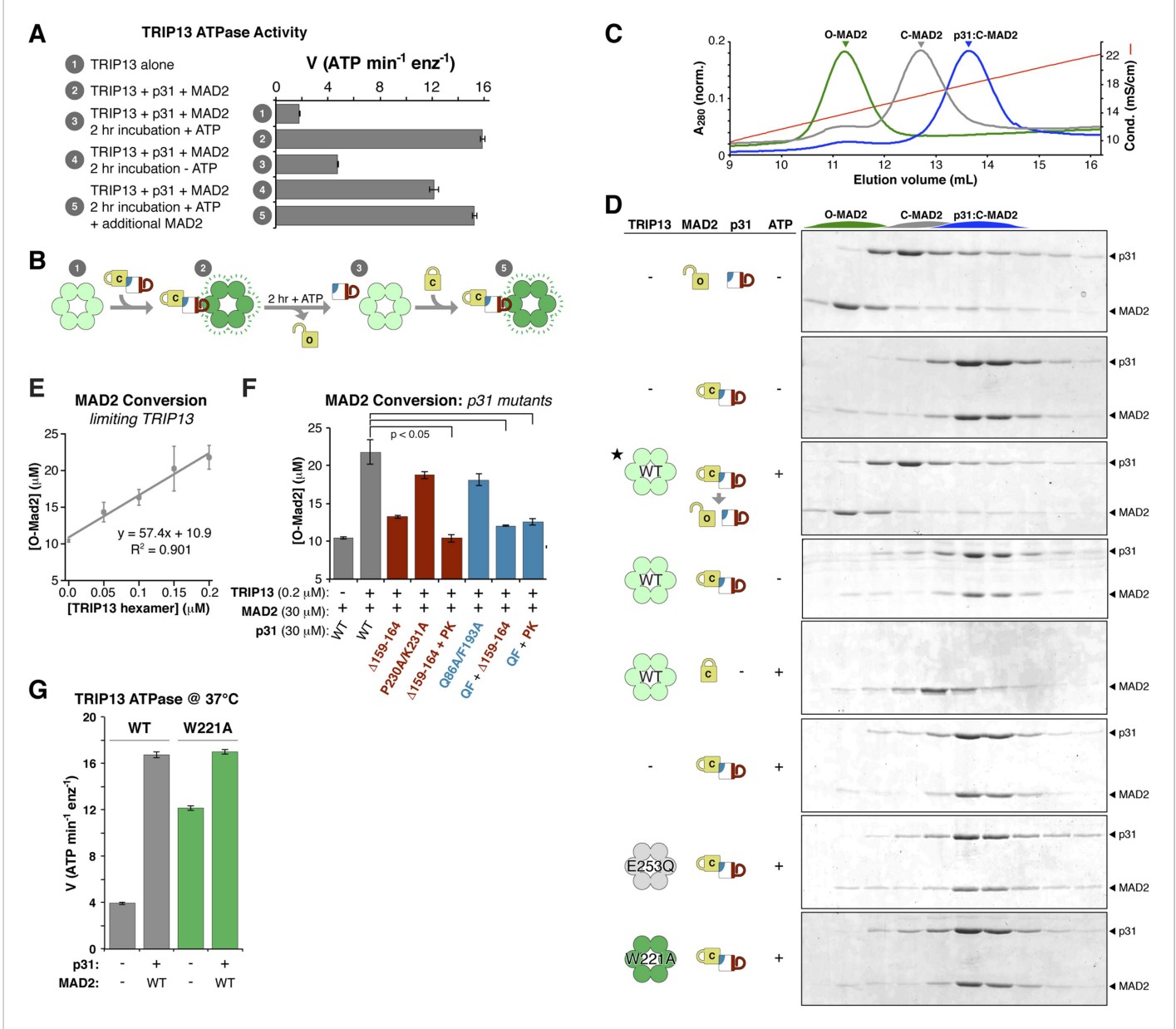

**Figure 7**. TRIP13 converts C-MAD2 to O-MAD2. (**A**) Stimulation of TRIP13 ATPase activity by p31(comet) + MAD2, before (samples 1–2) or after (samples 3–5) a 2-hr pre-incubation period. All proteins were at 4 µM (sixfold molar excess of substrate:TRIP13 hexamer). For sample 5, an additional 4 µM MAD2 was added after the pre-incubation period. (**B**) Schematic illustrating results from (**A**) in terms of complex formation and TRIP13 stimulation. (**C**) Anion-exchange elution profiles for O-MAD2 (green), C-MAD2 (gray), and the p31(comet):C-MAD2 complex (blue). (**D**) p31(comet) and MAD2 in anion-exchange fractions from the indicated pre-incubated reaction mixtures. p31(comet) and MAD2 were at 30 µM, and TRIP13 at 5 µM (hexamer concentration) except for starred sample (third from top), where TRIP13 was at 1.25 µM. The observed C-MAD2 to O-MAD2 conversion requires active TRIP13, ATP, and p31(comet). Neither TRIP13$^{E253Q}$ nor TRIP13$^{W221A}$ supported MAD2 conversion. At equilibrium, MAD2$^{R133A}$ is predominantly in the C-MAD2 state (**Figure 7—figure supplement 1**), further supporting that the observed C-MAD2 to O-MAD2 conversion is an active process. (**E**) Rate of TRIP13-mediated C-MAD2 to O-MAD2 conversion in limiting TRIP13. Reactions with 30 µM p31(comet) + MAD2$^{R133A}$ (initially ~10 µM O-MAD2 and ~20 µM C-MAD2) were incubated at 37°C for 30 min with the indicated amounts of TRIP13, and separated by ion-exchange as in (**D**). SDS-PAGE band intensities were quantified, converted to [O-MAD2], and plotted. Linear regression fitting indicates a rate of $57.4 \pm 6.7$ MAD2 conversions in 30 min per TRIP13 hexamer, or $\sim 1.9 \pm 0.2$ min$^{-1}$ (**F**) TRIP13-mediated MAD2 conversion in the presence of p31(comet) mutants. The high concentration of p31(comet) + MAD2 in this assay (30 µM)

*Figure 7. continued on next page*

*Figure 7. Continued*

allowed single mutants to support limited MAD2 conversion, but mutant combinations effectively eliminated MAD2 conversion. PK: P230A/K231A; QF: Q86A/F193A. (**G**) ATPase activity of *M. musculus* TRIP13 at 37°C (all other ATPase assays were performed at 27°C).

The following figure supplement is available for figure 7:

**Figure supplement 1**. MAD2$^{R133A}$ is predominantly in the C-MAD2 state at equilibrium.

# Discussion

Our data show that Pch2/TRIP13 is a AAA+ ATPase with structural and mechanistic properties similar to both the 'classic remodelers' and the bacterial protein unfoldase ClpX. First, the PCH-2 hexamer architecture shows close mechanistic parallels with ClpX, suggesting a shared mechanism for nucleotide-dependent conformational changes driving pore loop motions and substrate remodeling. Our structural and biochemical data suggest that Pch2/TRIP13 engages its substrates within the hexamer pore, and undergoes coordinated ATP hydrolysis-coupled conformational changes to mediate substrate unfolding. In contrast to ClpX, however, Pch2/TRIP13 does not completely unfold its HORMA domain protein substrates. Rather, we have shown that TRIP13 catalyzes a much more subtle structural change, converting closed MAD2 to its open state. We propose that TRIP13 specifically unfolds the C-terminal safety belt region of MAD2, then allows it to refold into the open state. Given the mechanistic similarities between Pch2/TRIP13 and the processive unfoldase ClpX, however, an obvious question is how unfolding by Pch2/TRIP13 is controlled to achieve HORMA domain conformational conversion instead of complete unfolding.

The answer to this question may lie in the second aspect of Pch2/TRIP13's hybrid nature: its mode of substrate recognition, which is mediated by an NTD related to a family of 'classic remodelers' including NSF, p97, and PEX1. Importantly, as in NSF and p97, substrate recognition by Pch2/TRIP13 is indirect with p31(comet) acting as an adapter to deliver MAD2 to TRIP13. p31(comet) binds specifically to C-MAD2, meaning that once TRIP13 engages and unfolds the MAD2 safety belt, p31(comet) would cease to bind MAD2. This could destabilize the ternary complex, releasing partially-unfolded MAD2 and allowing its re-folding into the open conformation. As mentioned above, MAD2 safety belt unfolding could occur in a processive manner accompanied by multiple rounds of ATP hydrolysis by TRIP13, or could occur similarly to NSF, where recent work has indicated a single-step 'spring-loaded' mechanism for SNARE complex disassembly (*Ryu et al., 2015*). In the SAC, we propose that p31(comet) and TRIP13 catalyze a two-step MCC disassembly mechanism to inactivate the SAC (*Figure 8A,B*). First, p31(comet) displaces BUBR1 from MAD2, potentially causing its dissociation from MAD2:CDC20. The resulting p31(comet):MAD2:CDC20 complex is then recognized by TRIP13, which converts C-MAD2 to O-MAD2, thus disrupting binding to both p31(comet) and CDC20, and also preventing MCC re-assembly.

Essentially all free MAD2 in HeLa cells is in the open state (*Luo et al., 2004*), but prior work (*Luo et al., 2004*) and our own analysis (*Figure 7—figure supplement 1*) indicates that while the two conformations are relatively stable at 4°C, at physiological temperatures essentially all O-MAD2 spontaneously converts to C-MAD2 within several hours. These data strongly suggest that cellular factors actively maintain MAD2 in the open state. We propose that a major role for TRIP13 and p31(comet) may be to counteract spontaneous O-MAD2 to C-MAD2 conversion, thus guarding against improperly-timed MCC assembly (which can occur outside mitosis given a supply of soluble C-MAD2 [*Tipton et al., 2011a*]) and also ensuring a sufficient supply of O-MAD2 for SAC activation in prometaphase. An important remaining question is how the competing pathways for MCC assembly and disassembly are balanced and regulated throughout the cell cycle: do p31(comet) and TRIP13 constantly disassemble MCC at a low level during prometaphase and metaphase, with this activity becoming dominant only after new MCC assembly is ceased, or is the activity of p31(comet) and TRIP13 suppressed during metaphase by additional mechanisms? Recently it was shown that human p31(comet) is phosphorylated specifically in mitosis, and that phosphorylation lowers the affinity of p31(comet) for MAD2 (*Date et al., 2014*). While the phosphorylated residue (Ser102) is

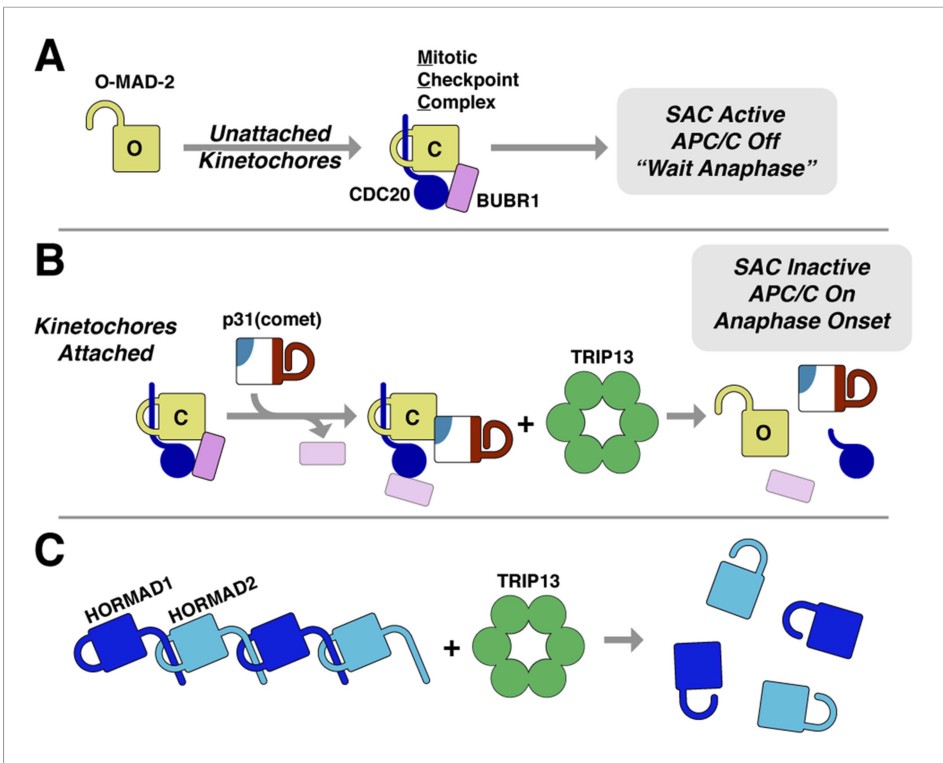

**Figure 8**. Model for SAC inactivation by p31(comet) and TRIP13. (**A**) Unattached kinetochores catalyze the assembly of the mitotic checkpoint complex (MCC) through the conversion of O-MAD2 to C-MAD2 and assembly with CDC20 (blue), BUBR1 (pink), and BUB3 (not shown). (**B**) After kinetochore-microtubule attachment, MCC assembly is halted. p31(comet) binds existing MCC and displaces BUBR1, then delivers C-MAD2:CDC20 to TRIP13 for conformational conversion and disassembly. The CDC20:BUBR1 interaction may be disrupted directly by p31(comet) or at a later point. (**C**) Scheme for TRIP13-mediated disassembly of HORMAD oligomers (blue) in meiosis. It is unknown whether HORMADs possess an open state analogous to O-MAD2. See *Figure 8—figure supplement 1* for structures of O-MAD2, C-MAD2, C-HORMAD, and p31(comet) showing the safety-belt conformation in each state.

The following figure supplement is available for figure 8:

**Figure supplement 1**. HORMA domain structures in open and closed conformation.

not universally conserved in p31(comet) orthologs, this result nonetheless represents one potential mechanism for suppressing TRIP13-mediated MCC disassembly specifically during mitosis.

The ability of TRIP13 to disengage the safety-belt motif of MAD2 strongly suggests a parallel mechanism for its remodeling/removal of HORMAD proteins along chromosomes in meiotic prophase (*Figure 8C*). We have previously shown that the meiotic HORMADs assemble into hierarchical head-to-tail complexes through safety-belt interactions, and that these interactions are crucial for meiotic DNA break formation, inter-homolog recombination, and chromosome segregation (*Kim et al., 2014*). We have been unable, however, to detect direct interactions between PCH-2/TRIP13 and their putative HORMAD substrates, nor do these proteins stimulate PCH-2/TRIP13 ATPase activity (not shown). Thus, how the enzyme recognizes HORMAD complexes, whether a p31(comet)-like adapter is needed for this recognition, and what signals coordinate crossover formation with HORMAD complex remodeling and removal, remain important open questions. Finally, given the additional association of human *TRIP13* with a number of cancer types (*Larkin et al., 2012*; *van Kester et al., 2012*; *Banerjee et al., 2014*), addressing the fundamental mechanistic questions regarding how this enzyme recognizes and remodels its substrates will be important for understanding TRIP13's multiple roles in human health and disease.

## Materials and methods

### Sequence analysis

For sequence analysis of AAA+ ATPases, isolated AAA+ regions (large plus small domains, isolated D1 domain for p97/Cdc48 and NSF) were aligned with MAFFT (*Katoh and Standley, 2013*), a phylogenetic tree was constructed in JalView (*Waterhouse et al., 2009*), and the tree was visualized with Dendroscope (*Huson and Scornavacca, 2012*).

### Protein expression and purification

Full-length *C. elegans* PCH-2 and *M. musculus* TRIP13 were cloned from cDNA into a bacterial expression vector with an N-terminal TEV protease-cleavable His$_6$ tag. Mutant constructs were generated by PCR-based mutagenesis: *C. elegans* PCH-2 ΔNTD consisted of residues 100–424, and the 'Pore loop AG' mutant replaced residues 218–227 with the protein sequence 'AGAAGAAAGA'. All PCH-2/TRIP13 mutants used for activity assays were expressed at levels similar to the wild-type proteins and migrated equivalently on a size-exclusion column, indicating that they are soluble and folded. The Walker A motif mutant K185Q of both PCH-2 or TRIP13, and the ΔNTD mutant of TRIP13, were not solubly expressed, precluding their analysis. For *C. elegans* MAD-2 (MDF-2) and CMT-1 (C41D11.5), and *M. musculus* MAD2 and p31(comet), full-length proteins were cloned from cDNA into a bacterial expression vector with an N-terminal TEV protease-cleavable His$_6$ tag. Mutant constructs were generated by PCR-based mutagenesis: *M. musculus* MAD2 'loopless' replaced residues 109–117 with the protein sequence 'GSG' as in (*Mapelli et al., 2007*) (see *Figure 8—figure supplement 1A*). For identification of highly conserved surface residues in p31(comet), 226 animal/plant p31(comet) sequences were aligned with MAFFT (*Katoh and Standley, 2013*) and conservation was mapped on the structure of p31(comet) bound to MAD2 (*Yang et al., 2007*) using the CONSURF server (*Ashkenazy et al., 2010*). All mutant constructs were generated by PCR-based mutagenesis. All mutant constructs used here (p31(comet), MAD2, and TRIP13) expressed at levels similar to wild-type, and migrated equivalently on a size-exclusion column (not shown), indicating that they were soluble and folded.

Proteins were expressed in *Escherichia coli* strain Rosetta 2 (DE3) pLysS (EMD Millipore, Billerica MA) at 20°C for 16 hr, then cells were harvested by centrifugation and resuspended in buffer A (25 mM Tris pH 7.5, 10% glycerol, 5 mM MgCl$_2$) plus 300 mM NaCl, 5 mM imidazole, and 5 mM β-mercaptoethanol. Protein was purified by Ni$^{2+}$-affinity (Ni-NTA agarose, Qiagen) then ion-exchange (Hitrap Q HP, GE Life Sciences, Piscataway NJ) chromatography. Tags were cleaved with TEV protease (*Tropea et al., 2009*), and cleaved protein was passed over a size exclusion column (Superdex 200, GE Life Sciences) in buffer A plus 300 mM NaCl and 1 mM dithiothreitol (DTT). Purified protein was concentrated by ultrafiltration (Amicon Ultra, EMD Millipore) to ~10 mg/ml and stored at 4°C. For selenomethionine derivatization of PCH-2, protein expression was carried out in M9 minimal media supplemented with amino acids plus selenomethionine prior to IPTG induction (*Van Duyne et al., 1993*), and proteins were exchanged into buffer containing 1 mM *tris*(2-carboxyethyl)phosphine (TCEP) after purification to maintain the selenomethionine residues in the reduced state.

For size-exclusion chromatography coupled multi-angle light scattering (SEC-MALS), proteins were separated on a Superdex 200 10/300 GL size exclusion column (GE Life Sciences), their light scattering and refractive index profiles collected by miniDAWN TREOS and Optilab T-rEX detectors (Wyatt Technology, Santa Barbara CA), respectively, and their molecular weights calculated using ASTRA v. 6 software (Wyatt Technology).

### Electron microscopy

For negative-stain EM, PCH-2 was passed over a size-exclusion column in EM buffer (buffer A without glycerol, and with added 1 mM DTT), then peak fractions were diluted to ~0.01 mg/ml in EM buffer with or without 1 mM ATP. Samples were applied onto freshly glow discharged carbon coated copper grids, reduced to a thin film by blotting, and a 2% solution of uranyl formate negative stain was then immediately applied to the grid and blotted off from the same side. The negative stain process was repeated 3 times. Data were acquired using a Tecnai F20 Twin transmission electron microscope (FEI, Hillsboro OR) operating at 200 kV. Images were automatically collected using the LEGINON system (*Suloway et al., 2005*). Images were recorded using a Tietz F416 4k × 4k pixel CMOS camera (TVIPS, Gauting, Germany).

Experimental data were processed by the APPION software package (*Lander et al., 2009*). The defoci were estimated using ctffind3 (*Mindell and Grigorieff, 2003*) and ACE2 (*Lander et al., 2009*) and CTF correction was done by phase flipping the whole micrograph. Particles were selected automatically in APPION using DogPicker (*Roseman, 2004*; *Voss et al., 2009*). After stack creation in APPION all datasets were prealigned and classified using 2-D maximum likelihood procedures and multivariate statistical analysis as implemented in XMIPP and IMAGIC (*Radermacher et al., 1986*; *van Heel et al., 1996*; *Sorzano et al., 2004*; *Scheres et al., 2008*). Resulting class averages were manually inspected and classes that represented noise or distorted particles were rejected. Final clustering was performed using XMIPP clustering 2D alignment (cl2d).

## Crystallization and structure solution

For crystallization, PCH-2 was exchanged into crystallization buffer (25 mM Tris pH 7.5, 5 mM MgCl$_2$, 200 mM NaCl, 1 mM tris(2-carboxyethyl)phosphine [TCEP]), either without added nucleotide (Apo) or with 1 mM ADP or non-hydrolyzable ATP analogs (ATP-γS or AMP-PCP). Regardless of added nucleotides, PCH-2 formed large prism-shaped crystals after mixing 1:1 with 100 mM sodium citrate pH 5.6, 200 mM ammonium sulfate, and 15% PEG 3350. Crystals were cryoprotected by the addition of 20% glycerol, and flash-frozen in liquid nitrogen. Diffraction data were collected at synchrotron sources (see *Table 1*), and processed with HKL2000 (*Otwinowski and Minor, 1997*) or XDS (*Kabsch, 2010*). All crystals were in space group C222$_1$, with one PCH-2 hexamer per asymmetric unit. The structure was determined using phases obtained from a single-wavelength anomalous diffraction (SAD) dataset from a crystal grown from selenomethionine-derivatized protein. Automated XDS → SHELX → PHENIX for the SAD dataset was performed by the RAPD data-processing pipeline at the Advanced Photon Source NE-CAT beamline 24ID-E (https://rapd. nec.aps.anl.gov/rapd). 46 selenomethionine sites were identified using SHELX as implemented in hkl2map (*Sheldrick, 2010*), then supplied to the AutoBuild module of PHENIX (*Terwilliger et al., 2009*; *Adams et al., 2010*), which located an addition 22 sites (for 68 total—66 sites would be expected for a hexamer of PCH-2, but in several cases two sites represented alternate rotamers for a single methionine residue), and calculated and refined phases using PHASER (*McCoy et al., 2007*) and RESOLVE (*Terwilliger et al., 2009*) (http://www.solve.lanl.gov). Initial sequence-threaded models of the large and small AAA domains were generated by the PHYRE2 server (*Kelley and Sternberg, 2009*) (http://www.sbg.bio.ic.ac.uk/phyre2) and manually placed to generate an initial model. Initial placement and refinement of this model allowed identification of twofold non-crystallographic symmetry, which was then used during early-stage map generation, model building, and refinement (final refinement was performed without non-crystallophic symmetry). Numerous rounds of manual rebuilding in Coot (*Emsley et al., 2010*) and refinement in phenix. refine (*Adams et al., 2010*) against a high resolution native dataset resulted in improved maps, allowing us to manually build the NTDs. Data were highly anisotropic, showing significantly lower intensity (I/σ) and half–set correlation (CC$_{1/2}$) (*Karplus and Diederichs, 2012*) along the c* axis than along a* and b* (*Table 1* and *Figure 2—figure supplement 1A*). For refinement, the high-resolution native dataset processed to 2.3 Å with XDS was submitted to the UCLA Diffraction Anisotropy Server (*Strong et al., 2006*) (http://services.mbi.ucla.edu/anisoscale/) for application of anisotropic cutoffs (2.3 Å along a* and b*, 3.2 Å along c*). The final model consists of six PCH-2 monomers, with a total of 2214 residues modeled out of 2544 (6 × 424 residues); the model displays good geometry with 98.05% of residues in favored, and 99.77% of residues in allowed Ramachandran space (*Table 1*). All crystallographic software was installed and maintained through the SBGrid program (*Morin et al., 2013*).

## Protein–protein interaction assays

For yeast two-hybrid analysis, full-length sequences for *M. musculus* TRIP13, MAD2, and p31(comet) (wild-type and mutants) were cloned into pGADT7 (Gal4 activation domain fusion: 'AD') and pBridge (Gal4 DNA binding domain fusion: 'BD') vectors (Clontech Laboratories, Mountain View CA). Plasmids were transformed into AH109 and Y187 yeast strains, and transformants selected on CSM -Leu (pGADT7) or CSM -Trp (pBridge) media. Haploid strains were mated overnight at room temperature, and diploids were selected on CSM -Leu-Trp media. Diploids were then patched onto CSM -Leu-Trp-His (low stringency; shown in *Figure 5B–C*) or CSM -Leu-Trp-His-Ade (high stringency; not shown, results consistent with low-stringency results) media, grown 3 days at 30°C, and imaged. For yeast

three-hybrid analysis, p31(comet) was cloned into multiple cloning site #2 of pBridge to express the untagged protein alongside the BD- and AD-fusion proteins.

For $Ni^{2+}$ pulldown assays, 300 picomoles (9.3 µg) $His_6$-tagged p31(comet) was mixed with 450 picomoles untagged MAD2 (10.7 µg) in 50 µl binding buffer (20 mM Tris-HCl pH 7.5 or 8.5, 200 mM NaCl, 20 mM imidazole, 1 mM β-mercaptoethanol, 5% glycerol, 0.1% NP-40), incubated 60 min 20°C, then 'load' samples (5 µl, 10%) were removed and samples were mixed with Ni-NTA magnetic beads (10 µl 5% suspension, Qiagen, Hilden, Germany) for 20 min 20°C. Samples were washed 3× with 1 ml binding buffer, then 25 µl SDS-PAGE loading buffer was added, samples were boiled, run on 12.5% SDS-PAGE gels and imaged by Coomassie staining.

For size-exclusion chromatography analysis of TRIP13 plus p31(comet):MAD2, equimolar amounts (10 nanomoles) of $TRIP13^{E253Q}$, p31(comet), and $C-MAD2^{R133A}$ were mixed in 300 µl total volume of gel filtration buffer (20 mM Tris-HCl pH 7.5, 300 mM NaCl, 10% Glycerol, 1 mM DTT) plus 2 mM ATP, and incubated on ice for 30 min before application on a size exclusion column (Superdex 200 Increase 10/300 GL, GE Life Sciences) in gel filtration buffer plus 0.1 mM ATP. For SEC-MALS analysis of selected complexes, the same protocol was followed except for addition of nucleotide to the column running buffer.

## ATPase and MAD2 conversion assays

For ATPase assays with *C. elegans* PCH-2, optimal basal ATPase rates were obtained from protein treated during the $Ni^{2+}$-affinity purification step with 0.8 M urea, which removes the two ADP molecules bound to each hexamer (as determined by UV absorbance; not shown), followed by addition of 50 mM ammonium sulfate to all subsequent purification steps (necessary for protein stability after ADP removal).

ATPase activity was determined at 27°C (except where indicated) using an enzyme-coupled assay (*Nørby, 1988*) adapted for a microplate reader (*Kiianitsa et al., 2003*). 100 µl reactions contained assay buffer (25 mM Tris-HCl at pH 7.5 or 8.5 (see below), 200 mM NaCl, 10 mM $MgCl_2$, 1 mM DTT, 5% glycerol plus 2 mM ATP, 3 mM phosphoenolpyruvate, 20 U/ml lactate dehydrogenase (Sigma Aldrich, St. Louis MO), 20 U/ml pyruvate kinase (Sigma Aldrich), and 0.3 mM NADH. All PCH-2/TRIP13 constructs showed a strong dependence on pH with almost undetectable activity at pH 7.0 and below, and full activity at pH 8.5–9.5. For assays measuring stimulation of ATPase activity by p31(comet) and MAD2, assays were performed at pH 7.5 where basal activity was lower but substrate stimulation was robust. The assayed concentration of each PCH-2/TRIP13 construct was adjusted between 0.5 and 20 µM monomer, for the most accurate measurement of ATPase activity, depending on the assay. For TRIP13, which adopts multiple oligomeric states in solution, we verified that the ATP hydrolysis rate varies linearly between 0.625 and 10 µM TRIP13 (monomer concentration) at both pH 7.5 and 8.5, indicating that the protein is predominantly hexameric (and thus fully active) in our ATPase assay conditions (data not shown). Unless otherwise indicated, TRIP13 was equimolar with added p31(comet) and MAD2 (judging by TRIP13 monomer concentration). The decline of NADH absorbance at 340 nm was measured using a TECAN (Mannedorf, Switzerland) Infinite M1000 spectrophotometer in 384-well microplates. NADH oxidation rate was calculated from a linear fit to each time course and converted to ATP hydrolysis rates. For calculation of $K_m$ and $k_{cat}$, sampled were performed in triplicate with ATP concentration varying from 20 µM (lower bound for measuring decline in NADH absorbance) to 1 mM, and data were fit to the Michaelis-Menten equation ($Y = (V_{max} \times X)/(K_m + X)$) using PRISM v. 6 (GraphPad Software, La Jolla CA).

For samples measuring TRIP13 ATPase activity after pre-incubation, proteins (4 µM concentration for all proteins) were pre-incubated for 2 hr at 20°C in assay buffer with or without ATP as above. Samples were then passed through a desalting spin column (Zeba-Spin, Thermo Scientific, Waltham MA) to remove remaining ATP and hydrolyzed ADP. Fresh ATP and coupled-assay master mix were then added and ATP hydrolysis measured as above.

To examine MAD2 conformational conversion by TRIP13, separately purified C-MAD2 monomer (R133A mutant), p31(comet), and TRIP13 were incubated at 20°C for 2 hr at 30 µM concentration (sixfold molar excess of p31(comet) and MAD2 to TRIP13 hexamers), in ATPase assay buffer (pH 7.5) with or without 2 mM ATP (166 µl reaction volume). Samples were diluted to 50 mM NaCl by the addition of buffer without NaCl, then loaded onto a 1 ml HiTrap Q HP column (GE Life Sciences) and eluted with a gradient to 400 mM NaCl. Fractions were collected, analyzed by SDS-PAGE and visualized by Coomassie staining.

For measurement of MAD2 conversion rate by TRIP13 (*Figure 7E,F*), pre-incubations were performed for 30 min at 37℃ with 30 µM p31(comet), 30 µM MAD2 enriched for C-MAD2 (approximately 10 µM O-MAD2 and 20 µM C-MAD2), and the indicated amounts of TRIP13. Samples were separated by ion-exchange chromatography, and quantitation of Coomassie blue-stained SDS-PAGE bands was performed. Background-subtracted relative intensities of O-MAD2 vs C-MAD2 (lanes 6–7) was performed using ImageJ (*Schneider et al., 2012*), and ratios were converted to quantities based on the total (MAD2) of 30 µM. (O-MAD2) was plotted vs (TRIP13 hexamer) in the range of (TRIP13) where the reaction was not saturated (up to 0.2 µM TRIP13 hexamer). Linear regression fitting was performed with PRISM v. 6 (GraphPad Software).

## Acknowledgements

We thank D Cleveland, A Desai, J Han, A Shiau, J Berger, and members of the Corbett lab for helpful discussions. We thank APS NE-CAT (supported by NIH GM103403) and SSRL (supported by the DOE Office of Biological and Environmental Research and NIH P41GM103393) staff for assistance with x-ray data collection, B Carragher, C Potter, J Al-Bassam and D Lyumkis for assistance with EM data processing, S Dallakyen and A Herold for technical support, and J Anzola for assistance with ATPase assays. EM data collection and analysis was performed with the help of The National Resource for Automated Molecular Microscopy, which is supported by the National Institutes of Health (GM103310). KDC acknowledges support from the Ludwig Institute for Cancer Research, the Sidney Kimmel Foundation Kimmel Scholars program, The March of Dimes Foundation (Research Grant FY14-251), and the National Institutes of Health (R01GM104141).

## Additional information

### Funding

| Funder | Grant reference | Author |
|---|---|---|
| National Institutes of Health (NIH) | R01GM104141 | Qiaozhen Ye, Scott C Rosenberg, Tiffany Y Su, Kevin D Corbett |
| March of Dimes Foundation (March of Dimes Births Defect Foundation) | FY14-251 | Qiaozhen Ye, Scott C Rosenberg, Kevin D Corbett |
| Sidney Kimmel Foundation for Cancer Research | Ludwig Institute for Cancer Research | Kevin D Corbett |
| National Institutes of Health (NIH) | GM103310 | Arne Moeller, Jeffrey A Speir |
| U.S. Department of Energy (Department of Energy) | | Kevin D Corbett |
| National Institutes of Health (NIH) | P41GM103393 | Kevin D Corbett |
| National Institutes of Health (NIH) | GM103403 | Kevin D Corbett |

The funders had no role in study design, data collection and interpretation, or the decision to submit the work for publication.

### Author contributions

QY, Planned the experiments, Determined the PCH-2 crystal structure, Performed all biochemical assays, Conception and design, Acquisition of data, Analysis and interpretation of data, Drafting or revising the article; SCR, Performed yeast two-hybrid assays, Acquisition of data; AM, JAS, Collected, processed, and analyzed EM data, Acquisition of data, Analysis and interpretation of data; TYS, Purified proteins and performed affinity pull-down assays, Acquisition of data; KDC, Planned the experiments, Analyzed EM data, Wrote the paper, Conception and design, Acquisition of data, Analysis and interpretation of data, Drafting or revising the article

## Author ORCIDs

Kevin D Corbett, http://orcid.org/0000-0001-5854-2388

# Additional files

## Major datasets

The following dataset was generated:

| Author(s) | Year | Dataset title | Dataset ID and/or URL | Database, license, and accessibility information |
|---|---|---|---|---|
| Ye Q, Corbett KD | 2015 | Structure of C. elegans PCH-2 | http://www.pdb.org/pdb/explore/explore.do?structureId=4XGU | Publicly available at RCSB Protein Data Bank (4XGU). |

The following previously published datasets were used:

| Author(s) | Year | Dataset title | Dataset ID and/or URL | Database, license, and accessibility information |
|---|---|---|---|---|
| Glynn SE, Martin A, Nager AR, Baker TA, Sauer RT | 2009 | Crystal structure of nucleotide-free hexameric ClpX | http://www.rcsb.org/pdb/explore/explore.do?structureId=3HTE | Publicly available at RCSB Protein Data Bank (3HTE). |
| Luo X, Fang G, Coldiron M, Lin Y, Yu H, Kirschner MW, Wagner G | 2000 | Solution structure of the spindle assembly checkpoint protein human Mad2 | http://www.rcsb.org/pdb/explore/explore.do?structureId=1DUJ | Publicly available at RCSB Protein Data Bank (1DUJ). |
| Luo X, Tang Z, Xia G, Wassmann K, Matsumoto T, Rizo J, Yu H | 2004 | The Mad2 spindle checkpoint protein possesses two distinct natively folded states | http://www.rcsb.org/pdb/explore/explore.do?structureId=1S2H | Publicly available at RCSB Protein Data Bank (1S2H). |
| Luo X, Tang Z, Rizo J, Yu H | 2002 | The Mad2 Spindle Checkpoint Protein Undergoes Similar Major Conformational Changes upon Binding to Either Mad1 or Cdc20 | http://www.rcsb.org/pdb/explore/explore.do?structureId=1KLQ | Publicly available at RCSB Protein Data Bank (1KLQ). |
| Kim Y, Rosenberg SC, Kugel CL, Kostow N, Rog O, Davydov V, Su TY, Dernburg AF, Corbett KD | 2014 | Structure of C. elegans HIM-3 bound to HTP-3 closure motif-4 | http://www.rcsb.org/pdb/explore/explore.do?structureId=4TZJ | Publicly available at RCSB Protein Data Bank (4TZJ). |
| Yang M, Li B, Tomchick DR, Machius M, Rizo J, Yu H, Luo X | 2007 | Crystal structure of the Mad2/p31(comet)/Mad2-binding peptide ternary complex | http://www.rcsb.org/pdb/explore/explore.do?structureId=2QYF | Publicly available at RCSB Protein Data Bank (2QYF). |

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
