## [Decision Letter]

Thank you for sending your work entitled “TRIP13 is a protein-remodeling AAA+ ATPase that catalyzes MAD2 conformation switching” for consideration at *eLife*. Your article has been favorably evaluated by John Kuriyan (Senior editor) and three reviewers, one of whom is a member of our Board of Reviewing Editors.

The following individuals responsible for the peer review of your submission have agreed to reveal their identity: Axel Brunger (Reviewing editor); Andreas Martin (peer reviewer). A further reviewer remains anonymous.

The Reviewing editor and the other reviewers discussed their comments before we reached this decision, and the Reviewing editor has assembled the following comments to help you prepare a revised submission.

This manuscript describes a crystal structure of the TRIP13/Pch2 ATPase, along with negative stain electron microscopy microcopy (low resolution class averages), and the reconstitution in vitro of a crucial role of the ATPase, the conversion of C-MAD2 to O-MAD2. There are two interesting new findings: a pronounced type of asymmetry in the PCH-2 (TRIP13 homolog) AAA+ ATPase – two of the subunits in the hexametric arrangement are bound to sulfate, two are apparently apo, and two are bound to ADP, resulting in large conformational differences between the “small” subdomain of the AAA+ module.

A second important result is the finding that TRIP13 can induce a conformational change in its substrate MAD-2 (“closed” to “open”), via the adapter protein p31. This function of TRIP13 may play a role during mitotic and meiotic chromosome segregation. In summary, this work provides new insights into the structure and function of TRIP13/PCH-2. However, there were some concerns about the interpretation and generality of the asymmetry observed in the crystal structure, and about the ATP hydrolysis experiments, as well as numerous suggestions on discussion of results from other groups on other systems.

*Reviewer #1*:

1) Figure 1 implies that AAA domains A and D are in the apo state. Is this correct? Are there conformational differences between apo and ADP bound states?

2) Is it certain that A and D are apo? Please provide simulated annealing omit maps of the nucleotide binding regions of each of the six subunits.

3) Figure 2–figure supplement 2: The comparison to ClpX is problematic since in that structure all subunits were in the apo state.

4) Figure 4: does p31 interact with TRIP13 alone? In the case of NSF/aSNAP/SNARE, the adapter aSNAP does not interact with NSF alone.

5) The negative-stain EM studies are rather low resolution, and no 3D reconstruction was attempted. It should be made clear in the text that these results are rather tentative considering the lack of a 3D reconstruction and the lack of a resolution estimate.

6) There are too many supplementary figures, making it hard to read this paper. *eLife* discourages use of supplementary figures unless they represent raw data or repeat data. The authors are encouraged to restructure all their figures into a well-organized set of main figures with minimal use of supplementary figures.

Other comments to consider:

*Reviewer #1*:

1) Reference to Cipriano et al.: although Cipriano et al. measured 60 ATPs, others measured 10 ATPs (Shah, N., Colbert, K. N., Enos, M. D., Herschlag, D., & Weis, W. I. (2015). Three αSNAP and 10 ATP Molecules are used in SNARE Complex Disassembly by N -ethylmaleimide-sensitive Factor (NSF). Journal of Biological Chemistry, 290, 2175-2188).

2) Figure 4–figure supplement 4: which nucleotide state was used?

3) Figure 7–figure supplement 7, panel A: there are two structures shown (open and closed conformations). Please provide the PDB IDs of both structures.

4) Table 1: the B-factors of the anisotropic scale factor tensor should be provided.

*Reviewer #2*:

1) Introduction: The references [19], [30], [35] should be, if not replaced, at least supplemented with the two references that actually described the safety belt of MAD2 as a dynamic structural element, i.e. Luo et al. Mol. Cell 2002 and Sironi et al. EMBO J. 2002 (not to be confused with Sironi et al. EMBO J 2001, see below).

2) Tipton et al. (2013; JBC 286) were the first authors to demonstrate the interaction of BubR1 with C-MAD2 that p31comet interferes with. The authors may elect to cite this paper alongside [7]. The other two cited papers, albeit excellent, do not demonstrate the point the authors are discussing, i.e. the separation of C- MAD2 from BubR1. The reason to make this point is that the authors cite literature demonstrating a role of p31comet in MCC disassembly in a somewhat arbitrary and selective manner: there are quite a few additional papers making the point in addition to [84] and it is immediately clear why the authors pick one but not the others.

3) Discussion, first paragraph: The authors adopt an interpretation of the relative stability of O-MAD2 and C-MAD2 that is based on the studies of [37]. There is a second interpretation, discussed by Mapelli et al. COSB 2007, in which O-MAD2 is seen as being more stable than empty (without a ligand) C-MAD2, but much less stable than ligand-bound C-MAD2. In this view, O-MAD2 is not a metastable state, but rather the stable state of MAD2 in the absence of a binding partner. The authors should consider that both the dimerization of MAD2 (even residual dimerization as in the R133A mutant) and the binding of C-MAD2 to p31comet (which they use in their assay) influence very dramatically the O-MAD2:C-MAD2 equilibrium, and that measuring the relative energy of these two states is therefore very complex. These considerations should probably reverberate also on the authors' discussion on the amount of ATP used for the conversion reaction. If a ligand (e.g. Cdc20) were bound to C-MAD2, the stability of C-MAD2 would be much higher and probably the amount of ATP required for the conversion would be correspondingly higher. My feeling is that the low ATP requirements reported by the authors are essentially due to the fact that the empty C-MAD2 substrate they use is much less stable than a C-MAD2 bound to a substrate, and that all the energy essentially goes in disrupting the considerable stabilization to the C-MAD2 conformation provided by p31comet.

*Reviewer #3*:

1) TRIP13/PCH-2 is presented as a new AAA+ family, based on the observation that its N-terminal domain resembles those of classic remodelers, whereas its AAA+ domain shows stronger similarities to the HCLR clade of enzymes. The authors repeatedly emphasize that classic remodelers have a diverged mechanism for ATP-driven conformational changes, primarily due to the lack of a sensor-II Arg. However, to my knowledge there is no existing literature or strong data yet indicating a fundamental difference in mechanisms. Even the exact role of the sensor-II Arg is still unclear. Studies on Hsp104 suggested that the sensor-II Arg in NBD2 provides some extra nucleotide-binding energy but does not act to sense differences between nucleotides (Hattendorf and Lindquist, PNAS 99, 2002). In contrast, a sensor-II Arg mutation in ClpX allows normal ATP- or ADP binding but apparently prevents sensing of the nucleotide state (Joshi et al., NSMB 11, 2004).

The authors also discuss the distinctly asymmetric hexamer architecture of PCH-2 as an indication for a structural mechanism that is distinct from “classic remodelers”. However, FtsH, a member of the classic clade of AAA+ enzymes, also crystallized as an asymmetric hexamer that resembled a dimer of trimers (Bieniossek et al., PNAS 103, 2006). There may be several different reasons why some AAA+ enzymes crystallize in 2-fold symmetric hexamers, like TRIP13/PCH-2 or ClpX, whereas others are more symmetric or have a continuous spiral staircase arrangement of subunits. As shown by EM studies in this manuscript, ATP binding can already significantly influence ring symmetry, and substrate binding in the central pore may induce additional changes in subunit arrangement or vertical registers. Even for ClpX, for which there are numerous structures and functional data both in bulk and at the single-molecule level, it remains unclear whether during substrate processing the hexamer maintains a 2-fold symmetry or adopts a conformation that contains a continuum of nucleotide states and a corresponding spiral staircase arrangement of pore-loops around the ring.

I therefore suggest that the authors focus on the obvious differences and similarities in structure and sequence between TRIP13/PCH-2 and the HCLR or classic clades (including the sensor-II motif and Arg finger), without speculating and claiming too much about potential differences in the mechanisms of ATP-hydrolysis coupled conformational changes.

Related to this, the authors state: “Mutation of the putative Sensor-2 residue R385 in PCH-2 also results in the loss of nucleotide binding, illustrating the importance of this residue and further supporting a ClpX-like mechanism…”.

As mentioned above, ClpX with a mutation of the sensor-II Arg still has near wild-type affinity for ATP and ADP, and the authors should correct or clarify their statement to prevent confusion.

2) Even though it is tempting and reasonable to interpret the PCH-2 crystal structure with respect to potential structural intermediates adopted by individual subunits during the ATPase cycle, the authors should more strongly consider and discuss that this is still “only” an empty/ADP-bound structure. As indicated by the presented negative-stain EM as well as several previous studies on related AAA+ enzymes, ATP binding to the hexamer may significantly affect subunit conformations and ring topology. I assume the authors unsuccessfully tried to solve a PCH-2 crystal structure in the presence of a non-hydrolyzable ATP analog or for the Walker-B mutant in the presence of ATP, and therefore turned to negative stain EM. It would be worth mentioning that the crystallization of the ATP-bound state was unsuccessful.

3) In their discussion of potential nucleotide-driven conformational changes of PCH-2, pore-loop movements, and the strong similarities to the ClpX motor (in the subsection headed “Nucleotide-driven conformational changes in the PCH-2 hexamer” and Figure 2) the authors may include a recent paper by the Sauer group, indicating highly coordinated gripping and conformational changes of subunits in the ClpX hexamer (Iosefson et al., Nat Chem Bio 11, 2015). This is particularly relevant for the model of axial pore loop movements presented in Figure 2.

4) The authors performed Michaelis-Menten analyses and calculated *k*_*cat*_ for ATP hydrolysis by PCH-2 and TRIP13 (Figure 3). In the legend for Figure 1–figure supplement 2 it is mentioned that TRIP13 adopts a mixture of oligomeric states that shifts to higher molecular weight oligomers upon addition of ATP. For an accurate calculation of *k*_*cat*_ the question remains: how much TRIP13 was present in hexameric versus other states? It may be useful to measure the dependence of ATP hydrolysis on TRIP13 concentration to get an idea about the Kd for hexamerization and thereby increase confidence in the calculated *k*_*cat*_ value.

For the measurement of C-MAD2 substrate turnover the authors varied the TRIP13 concentration and observed a linear dependence (Figure 6), which suggests that the oligomeric state of TRIP13 did not change between 50 and 200 nM. However, it is possible that binding of p31/C-MAD2 to TRIP13 stabilizes the hexamer, especially because the single bound substrate-adaptor complex may bridge several TRIP13 subunits. It would therefore still be reasonable to assess the TRIP13-concentration dependence of ATP hydrolysis and get an idea about the extent of hexamers present under the experimental conditions used in the ATPase assay.

5) In the legend to Figure 1–figure supplement 2 the authors mention that the E253Q mutant of TRIP13 is predominantly hexameric both in the presence and the absence of ATP, suggesting that the mutation itself stabilizes the hexameric state. How did the authors determine that this mutant is indeed nucleotide-free when purified? Like other AAA+ enzymes TRIP13 may hold on to nucleotide during purification, even more so when it is hydrolysis-dead due to the Walker-B mutation.

6) It is a bit of a concern that the ATP-hydrolysis activity of PCH-2/TRIP13 is 90% inhibited at pH 7.5 compared to pH 8.5, and that the ATPase stimulation by p31/C-MAD2 can only be observed under these conditions. The authors should try to explain this behavior, as one would expect that the motor respond with a similar relative stimulation to the presence substrate in the central pore, independent of the pH and the corresponding basal ATPase rate. Does p31/C-MAD2 bind to TRIP13 with similar affinities at both pH 7.5 and 8.5? Currently there is no information in the figure legends or Materials and methods about the detailed buffer conditions used for SEC analyses of TRIP13/p31/MAD2 complex formation presented in Figure 4 and Figure 4–figure supplement 3 and 4.

The authors should for instance rule out that the observed ATPase stimulation by p31/C-MAD2 at pH 7.5 is simply due to a stabilization of the hexamer.

7) The authors show that only one p31/MAD2 complex binds to the TRIP13 hexamer, despite the presence of 6 N-terminal domains. A substrate delivery model is suggested in which p31 transiently interacts with the TRIP13 N-terminal domain before MAD2 directly binds in the central pore. However, currently this is largely speculation, and the authors should test this model by assessing the complex formation for the W221A pore loop mutant of TRIP13.

8) It is described in Materials and methods that “…optimal basal ATPase rates were obtained from protein treated during the Ni^2+^-affinity purification step with 0.8 M urea, which removes the two ADP molecules bound to each hexamer…”.

It is unclear to me why the co-purified ADP needed to be removed prior to ATPase measurements, during which ADP is constantly generated and quickly turned into ATP by the regeneration system. Is this stripping step necessary to more accurately measure the PCH-2 concentration by UV absorbance? Or does PCH-2 with 2 bound ADP represent a trapped, inactive form of the hexamer, which may also be prevented by purifying PCH-2 in the presence of ATP?

9) The ATPase measurements for the W221A mutant of TRIP13 showed absolutely no response to the presence of p31 and MAD2 at 27°C (Figure 5), but at 37°C there is a more than 30% increase in hydrolysis activity (Figure 6—figure supplement 1). The authors should try to explain those temperature-dependent differences in behavior.

10) In the Discussion section the authors discuss TRIP13's mode of substrate recognition, which seems to include an initial interaction of the p31 adaptor with the TRIP13 N-terminal domain, before the MAD2 substrate binds in the central pore. The authors state: “its mode of substrate recognition, which parallels not ClpX but a family of ‘classic remodelers’ including NSF, p97, and Pex1.” I am sure the Corbett group is aware that many ClpX substrates with N-terminal degradation signals initially interact with the ClpX N-terminal domain before they are transferred to the central pore, and ClpX adaptors also bind to the N-terminal domain for substrate delivery to the central pore. The overall process of substrate delivery to TRIP13 thus appears to strongly parallel the mechanisms of ClpX, even though there are differences in the architecture of the N-terminal domains. The authors should clarify or correct their statement to prevent confusion.

11) There is a mistake in a figure reference in the last paragraph of the subsection headed “p31(comet) functions as an adapter between MAD2 and TRIP13”. The sentence “Semi-quantitative analysis of Coomassie-stained gels, and light-scattering based molecular-weight measurements on this peak, revealed that for each TRIP13E235Q hexamer, about 1 copy of each protein was shifted into the TRIP13 peak (Figure 4, Figure 4–figure supplement 4)…” should reference Figure 4, not 4D.

---

## [Author Response]

Reviewer #1:

*1)*
Figure 1
*implies that AAA domains A and D are in the apo state. Is this correct? Are there conformational differences between apo and ADP bound states*?

Chains A and D are both bound to a SO_4_^-^ ion; we apologize for the confusion. We have restructured Figure 1 (now 2A) to include an explicit note of nucleotide/SO_4_^-^ binding in each active site. We now also provide views of all six active sites (with *F*_*o*_*-F*_*c*_ simulated annealing omit maps) in Figure 2—figure supplement 1. As for conformation, while chains A/D are bound to SO_4_^-^ and chains B/E are bound to ADP, their conformations are essentially identical, with an overall Cα r.m.s.d. of about 2 Å between them. Chains C/F are also bound to SO_4_^-^ but show a 70° rotation of the small AAA domain relative to the large AAA domain.

*2) Is it certain that A and D are apo? Please provide simulated annealing omit maps of the nucleotide binding regions of each of the six subunits*.

We performed simulated annealing refinement (torsion angle refinement, starting temperature 2500 K) in phenix.refine on a model with all nucleotides, SO_4_^-^ ions, and water atoms removed. The results show clear density for SO_4_^-^ ion in chains A, C, D and F, and ADP in chains B and E. The final *F*_*o*_*-F*_*c*_ maps from the SA refinement can be seen in Figure 2—figure supplement 1.

*3) Figure 2–figure supplement 2: The comparison to ClpX is problematic since in that structure all subunits were in the apo state*.

It is true that the ClpX structure cited in our original submission is in the apo state, but later structural work from the Sauer group showed that this hexamer conformation is compatible with nucleotide binding, with ATPγS bound to the active sites of the four “closed” (referred to by the Sauer group as nucleotide-“loadable”) chains: A, B, D, and E (PDB 4I81, Stinson et al. Cell 2013). The ATPγS-bound ClpX structure supports the idea that in both ClpX and in PCH-2, the closed/loadable conformation represents a nucleotide-bound like state, and the open/unloadable conformation represents an Apo state. Thus, while the actual nucleotide-binding state of both ClpX structures differs from our PCH-2 Apo/ADP structure, the information these structures possess regarding the likely motions that occur during nucleotide binding/hydrolysis/release is comparable.

To make this point clearer in the text, we have added a reference and a discussion of the later work showing the ClpX ATPγS structure (in the subsection headed “Nucleotide-driven conformational changes in the PCH-2 hexamer”). We have not altered Figure 3 (was Figure 2–figure supplement 2) to include the ClpX ATPγS structure since in that structure, the pore loops were not as well-ordered as in the Apo structure.

*4)*
Figure 4*: does p31 interact with TRIP13 alone? In the case of NSF/aSNAP/SNARE, the adapter aSNAP does not interact with NSF alone*.

p31(comet) does indeed interact with TRIP13 on its own, albeit more weakly than when it is bound to C-MAD2. This can be seen in our (and others’) yeast two-hybrid results showing a binary interaction in the absence of MAD2, and our size-exclusion chromatography analysis (Figure 5—figure supplement 1) showing that a small amount of p31(comet) shifts into the TRIP13 hexamer peak in the absence of MAD2. We propose that p31(comet) is able to recognize and bind the TRIP13 NTD without MAD2, but that MAD2 engaging within the TRIP13 hexamer pore further stabilizes the interaction.

*5) The negative-stain EM studies are rather low resolution, and no 3D reconstruction was attempted. It should be made clear in the text that these results are rather tentative considering the lack of a 3D reconstruction and the lack of a resolution estimate*.

We have made it clearer in the text that the EM analysis is very limited, given the dominance of top-down views in the data, and the lack of a 3D reconstruction. Nonetheless, we feel that these data are worth including as an indication that the PCH-2 hexamer can undergo what appear to major conformational changes upon nucleotide binding.

*6) There are too many supplementary figures, making it hard to read this paper.* eLife *discourages use of supplementary figures unless they represent raw data or repeat data. The authors are encouraged to restructure all their figures into a well-organized set of main figures with minimal use of supplementary figures*.

We have significantly streamlined the figures to make the narrative easier to follow. There are now eight main figures, each with two or fewer figure supplements, and less overall repetition of data.

*Other comments to consider*:

Reviewer #1:

*1) Reference to Cipriano et al.: although Cipriano et al. measured 60 ATPs, others measured 10 ATPs (Shah, N., Colbert, K. N., Enos, M. D., Herschlag, D., & Weis, W. I. (2015). Three αSNAP and 10 ATP Molecules are used in SNARE Complex Disassembly by N -ethylmaleimide-sensitive Factor (NSF). Journal of Biological Chemistry, 290, 2175-2188)*.

We thank the reviewer for pointing out this reference; we have cited it alongside Cipriano et al. It does seem that in the Shah experiment, the higher observed efficiency may be partly due to the trimeric αSNAP construct used; nonetheless, the measured number of ∼10 ATPs per event likely represents maximally efficient NSF-mediated SNARE disassembly.

*2) Figure 4–figure supplement 4: which nucleotide state was used*?

In these experiments, proteins were mixed in the presence of 2 mM ATP (TRIP13^E253Q^ was used to both stabilize the TRIP13 hexamer and eliminate hydrolysis), then the size exclusion column was run in buffer lacking nucleotide (while the SEC-MALS runs were performed without nucleotide in the running buffer to maximize signal/noise; the more comprehensive set of SEC runs were performed with 0.1 mM ATP in the running buffer. The consistent results between these conditions indicate that TRIP13^E253Q^ is likely stably bound to ATP after pre-incubation). We realized that a description of these experiments had been omitted from the Methods section; this has been remedied in the revised version of the manuscript.

*3) Figure 7–figure supplement 7, panel A: there are two structures shown (open and closed conformations). Please provide the PDB IDs of both structures*.

In the original version of this figure, both O-MAD2 (loopless) and C-MAD2 were taken from the MAD2 “conformational dimer” structure (PDB ID 2V64, [40]). We have now altered this figure to use the first reported structure of MAD2 in each conformation, and the PDB ID and reference for each panel is noted in the figure legend (now Figure 8—figure supplement 1). We also now include the structure of unliganded C-MAD2 in this figure.

*4)*
Table 1*: the B-factors of the anisotropic scale factor tensor should be provided*.

We have added the requested data to Table 1, along with the CC_1/2_ scores for each direction at the maximum resolution used for the elliptical data cutoffs.

Reviewer #2:

*1) Introduction: The references*
[19], [30], [35]
*should be, if not replaced, at least supplemented with the two references that actually described the safety belt of MAD2 as a dynamic structural element, i.e. Luo et al. Mol. Cell 2002 and Sironi et al. EMBO J. 2002 (not to be confused with Sironi et al. EMBO J 2001, see below)*.

We thank the reviewer for pointing out this error, and have added the two noted references. We have also kept the citations of Hara and Kim, as they are examples of distinct HORMA domain protein families binding their partners in a manner equivalent to MAD2.

*2) Tipton et al. (2013; JBC 286) were the first authors to demonstrate the interaction of BubR1 with C-MAD2 that p31comet interferes with. The authors may elect to cite this paper alongside*
[7]*. The other two cited papers, albeit excellent, do not demonstrate the point the authors are discussing, i.e. the separation of C-MAD2 from BubR1. The reason to make this point is that the authors cite literature demonstrating a role of p31comet in MCC disassembly in a somewhat arbitrary and selective manner: there are quite a few additional papers making the point in addition to*
[84]
*and it is immediately clear why the authors pick one but not the others*.

The reviewer is absolutely correct that our original submission did not properly reference the p31(comet) literature, and did not properly credit the several groups that have contributed in this area. We have restructured this section of the Introduction, added the Tipton et al. (2011) JBC reference noted, and also added several other references from different groups that show p31(comet)’s importance for SAC inactivation.

*3) Discussion, first paragraph:. The authors adopt an interpretation of the relative stability of O-MAD2 and C-MAD2 that is based on the studies of*
[37]*. There is a second interpretation, discussed by Mapelli et al. COSB 2007, in which O-MAD2 is seen as being more stable than empty (without a ligand) C-MAD2, but much less stable than ligand-bound C-MAD2. In this view, O-MAD2 is not a metastable state, but rather the stable state of MAD2 in the absence of a binding partner. The authors should consider that both the dimerization of MAD2 (even residual dimerization as in the R133A mutant) and the binding of C-MAD2 to p31comet (which they use in their assay) influence very dramatically the O-MAD2:C-MAD2 equilibrium, and that measuring the relative energy of these two states is therefore very complex. These considerations should probably reverberate also on the authors' discussion on the amount of ATP used for the conversion reaction. If a ligand (e.g. Cdc20) were bound to C-MAD2, the stability of C-MAD2 would be much higher and probably the amount of ATP required for the conversion would be correspondingly higher. My feeling is that the low ATP requirements reported by the authors are essentially due to the fact that the empty C-MAD2 substrate they use is much less stable than a C-MAD2 bound to a substrate, and that all the energy essentially goes in disrupting the considerable stabilization to the C-MAD2 conformation provided by p31comet*.

The reviewer makes an excellent point, and we had not fully considered the complexity of the MAD2 energy landscape. To clarify the picture somewhat, we sought to determine the equilibrium levels of open and closed MAD2^R133A^ in vitro, using our ion-exchange chromatography assay. We performed these tests in the absence of any binding partners (which we agree do significantly stabilize the C-MAD2 conformation). We incubated purified O-MAD2 or C-MAD2 24 hours at 37°C, then separated them by ion-exchange and quantified their relative levels. We find that both samples converge to ∼15% O-MAD2 and ∼85% C-MAD2 (Figure 7—figure supplement 1). These results indicate that whatever the reason (dimerization, even residual dimerization as in MAD2^R133A^, could certainly promote conversion or at least stabilize the closed state), C-MAD2 is the preferred state in solution even in the absence of binding partners. Because of the complexity of this issue and our still-limited data, however, we have refrained from stating outright that C-MAD2 is more thermodynamically stable than O-MAD2.

Based on the above data and prior findings that O-MAD2 spontaneously converts to C-MAD2 ([37] and other references), we feel that our proposal that TRIP13 and p31(comet) may serve to maintain cellular MAD2 in the open state is still valid. Indeed, as noted by [37] and now explicitly noted in our discussion, essentially all free MAD2 in a HeLa cell extract is in the open state. Unless MAD2’s inherent conformational equilibrium is fundamentally different inside the cell versus outside, there will be a continuous (albeit slow) conversion of O-MAD2 to C-MAD2, which must be counteracted to avoid aberrant SAC activation, etc.

Apart from the above discussion, we agree with the reviewer that unliganded C-MAD2 is likely much less stable than liganded (or otherwise bound, e.g. to p31(comet) C-MAD2. We have added a sentence to the Results section (in the subsection headed “TRIP13 catalyzes the conversion of closed MAD2 to the open conformer”) pointing this out, and noting that conformational conversion of liganded C-MAD2 may take significantly more energy than what we measured.

Reviewer #3:

*1) TRIP13/PCH-2 is presented as a new AAA+ family, based on the observation that its N-terminal domain resembles those of classic remodelers, whereas its AAA+ domain shows stronger similarities to the HCLR clade of enzymes. The authors repeatedly emphasize that classic remodelers have a diverged mechanism for ATP-driven conformational changes, primarily due to the lack of a sensor-II Arg. However, to my knowledge there is no existing literature or strong data yet indicating a fundamental difference in mechanisms. Even the exact role of the sensor-II Arg is still unclear. Studies on Hsp104 suggested that the sensor-II Arg in NBD2 provides some extra nucleotide-binding energy but does not act to sense differences between nucleotides (Hattendorf and Lindquist, PNAS 99, 2002). In contrast, a sensor-II Arg mutation in ClpX allows normal ATP- or ADP binding but apparently prevents sensing of the nucleotide state (Joshi et al., NSMB 11, 2004)*.

We agree that the evidence for diverged mechanisms between the “classic remodelers” and other AAA+ proteins is not clear, so have re-worded the discussion of these points throughout the manuscript.

*The authors also discuss the distinctly asymmetric hexamer architecture of PCH-2 as an indication for a structural mechanism that is distinct from* “*classic remodelers*”*. However, FtsH, a member of the classic clade of AAA+ enzymes, also crystallized as an asymmetric hexamer that resembled a dimer of trimers (Bieniossek et al., PNAS 103, 2006). There may be several different reasons why some AAA+ enzymes crystallize in 2-fold symmetric hexamers, like TRIP13/PCH-2 or ClpX, whereas others are more symmetric or have a continuous spiral staircase arrangement of subunits. As shown by EM studies in this manuscript, ATP binding can already significantly influence ring symmetry, and substrate binding in the central pore may induce additional changes in subunit arrangement or vertical registers. Even for ClpX, for which there are numerous structures and functional data both in bulk and at the single-molecule level, it remains unclear whether during substrate processing the hexamer maintains a 2-fold symmetry or adopts a conformation that contains a continuum of nucleotide states and a corresponding spiral staircase arrangement of pore-loops around the ring*.

*I therefore suggest that the authors focus on the obvious differences and similarities in structure and sequence between TRIP13/PCH-2 and the HCLR or classic clades (including the sensor-II motif and Arg finger), without speculating and claiming too much about potential differences in the mechanisms of ATP-hydrolysis coupled conformational changes*.

We have restructured both the Results and Discussion sections on these points, with a clarified discussion of our findings and less speculation about the mechanistic uniqueness of Pch2/TRIP13.

*Related to this, the authors state:* “*Mutation of the putative Sensor-2 residue R385 in PCH-2 also results in the loss of nucleotide binding, illustrating the importance of this residue and further supporting a ClpX-like mechanism…*”.

*As mentioned above, ClpX with a mutation of the sensor-II Arg still has near wild-type affinity for ATP and ADP, and the authors should correct or clarify their statement to prevent confusion*.

We have removed the reference to ClpX in this sentence, and instead simply point out that the loss of nucleotide binding affinity in the R385 mutant strongly supports its identification as a Sensor-2 motif.

*2) Even though it is tempting and reasonable to interpret the PCH-2 crystal structure with respect to potential structural intermediates adopted by individual subunits during the ATPase cycle, the authors should more strongly consider and discuss that this is still* “*only*” *an empty/ADP-bound structure. As indicated by the presented negative-stain EM as well as several previous studies on related AAA+ enzymes, ATP binding to the hexamer may significantly affect subunit conformations and ring topology. I assume the authors unsuccessfully tried to solve a PCH-2 crystal structure in the presence of a non-hydrolyzable ATP analog or for the Walker-B mutant in the presence of ATP, and therefore turned to negative stain EM. It would be worth mentioning that the crystallization of the ATP-bound state was unsuccessful*.

While our EM analysis was conducted alongside crystallization trials, the reviewer is correct that we tried extensively to determine the structure of nucleotide-bound PCH-2, so far without success. Generally, we found that regardless of the added nucleotide, the resulting crystals adopted the same form as that determined here. After determining the Apo/ADP structure and then developing a method to strip out the bound nucleotide (see response to question 8 below), we were likely more successful in exchanging the nucleotide, but thus far we have not obtained crystals of protein treated in this manner. We now note our unsuccessful attempts to determine the PCH-2 structure in the presence of nucleotides in the subsection headed “Nucleotide-driven conformational changes in the PCH-2 hexamer” of the revised manuscript.

*3) In their discussion of potential nucleotide-driven conformational changes of PCH-2, pore-loop movements, and the strong similarities to the ClpX motor (in the subsection headed “Nucleotide-driven conformational changes in the PCH-2 hexamer” and*
Figure 2*) the authors may include a recent paper by the Sauer group, indicating highly coordinated gripping and conformational changes of subunits in the ClpX hexamer (Iosefson et al., Nat Chem Bio 11, 2015). This is particularly relevant for the model of axial pore loop movements presented in*
Figure 2.

We have included this reference as well.

*4) The authors performed Michaelis-Menten analyses and calculated* k_cat_
*for ATP hydrolysis by PCH-2 and TRIP13 (*Figure 3*). In the legend for Figure 1–figure supplement 2 it is mentioned that TRIP13 adopts a mixture of oligomeric states that shifts to higher molecular weight oligomers upon addition of ATP. For an accurate calculation of* k_cat_
*the question remains: how much TRIP13 was present in hexameric versus other states? It may be useful to measure the dependence of ATP hydrolysis on TRIP13 concentration to get an idea about the Kd for hexamerization and thereby increase confidence in the calculated* k_cat_
*value*.

*For the measurement of C-MAD2 substrate turnover the authors varied the TRIP13 concentration and observed a linear dependence (*Figure 6*), which suggests that the oligomeric state of TRIP13 did not change between 50 and 200 nM. However, it is possible that binding of p31/C-MAD2 to TRIP13 stabilizes the hexamer, especially because the single bound substrate-adaptor complex may bridge several TRIP13 subunits. It would therefore still be reasonable to assess the TRIP13-concentration dependence of ATP hydrolysis and get an idea about the extent of hexamers present under the experimental conditions used in the ATPase assay*.

As suggested by the reviewer, we measured the concentration dependence of TRIP13’s basal ATPase activity at both pH 7.5 and 8.5. The results, seen in Figure 9 (from triplicate measurements), indicate that within the TRIP13 concentration range we can reasonably test at each pH, the ATPase rate varies linearly with TRIP13 concentration (R^2^ values for both linear regression fits were 0.998). The slopes of these lines, which correspond to ATP’s hydrolyzed per minute per TRIP13 hexamer, are 3.36 at pH 7.5 and 8.64 at pH 8.5, close to the values reported in the manuscript. These data support the idea that wild-type TRIP13 is mostly hexameric in the ATPase assay conditions. We have noted this point in the Methods section (in the subsection headed “ATPase and MAD2 conversion assays”) as unpublished data.

Author response image 1.**DOI:**
http://dx.doi.org/10.7554/eLife.07367.022

*5) In the legend to Figure 1–figure supplement 2 the authors mention that the E253Q mutant of TRIP13 is predominantly hexameric both in the presence and the absence of ATP, suggesting that the mutation itself stabilizes the hexameric state. How did the authors determine that this mutant is indeed nucleotide-free when purified? Like other AAA+ enzymes TRIP13 may hold on to nucleotide during purification, even more so when it is hydrolysis-dead due to the Walker-B mutation*.

We were also concerned about this point, so have done some analysis to convince ourselves that TRIP13^E253Q^ is indeed “Apo” when purified. When we purify TRIP13^E253Q^ from *E. coli* (without adding nucleotides), the final size-exclusion step shows a major hexamer peak, and a minor monomer peak. During concentration and 4°C storage of the hexamer peak, extensive precipitation occurs, which can be eliminated by the addition of a nucleotide of our choice. This implies that the protein is not already bound to nucleotide when purified (or at least is not fully occupied), but is then stabilized upon the addition of nucleotides. Further, we have measured the UV absorbance spectrum for purified TRIP13^E253Q^ prior to nucleotide addition, and we obtain an A_260/280_ ratio of 0.651. From a theoretical calculation of TRIP13’s UV absorbance from the spectra of Trp, Tyr, and Phe, we obtain a theoretical A_260/280_ ratio of 0.682 for Apo protein, 0.786 for a TRIP13 hexamer bound to two nucleotides (as in PCH-2), and 0.977 for a hexamer bound to six nucleotides. Therefore, we are confident that TRIP13^E253Q^ is in the Apo state when purified.

A useful comparison can be made to the case of PCH-2, which does stably bind two ADP molecules per hexamer through the purification, except if treated with small amounts of denaturant during the initial Ni^2+^ affinity step (see response to point 8 below). We can clearly observe the two bound ADP’s, and their removal by denaturant treatment, by UV/Vis spectroscopy in that case.

*6) It is a bit of a concern that the ATP-hydrolysis activity of PCH-2/TRIP13 is 90% inhibited at pH 7.5 compared to pH 8.5, and that the ATPase stimulation by p31/C-MAD2 can only be observed under these conditions. The authors should try to explain this behavior, as one would expect that the motor respond with a similar relative stimulation to the presence substrate in the central pore, independent of the pH and the corresponding basal ATPase rate. Does p31/C-MAD2 bind to TRIP13 with similar affinities at both pH 7.5 and 8.5? Currently there is no information in the figure legends or Materials and methods about the detailed buffer conditions used for SEC analyses of TRIP13/p31/MAD2 complex formation presented in*
Figure 4
*and Figure 4–figure supplement 3 and 4*.

*The authors should for instance rule out that the observed ATPase stimulation by p31/C-MAD2 at pH 7.5 is simply due to a stabilization of the hexamer*.

We agree that the pH dependence of PCH-2/TRIP13 activity and stimulation by p31(comet):MAD2 is initially somewhat confusing. Is not the case, however, that there is no observed stimulation of PCH-2/TRIP13 by p31(comet):MAD2 at pH 8.5; only that the stimulation factor is not as high. For example, a TRIP13 hexamer hydrolyzes ∼2.5 ATP/min in the basal state versus ∼11 when stimulated at pH 7.5 (4X stimulation), versus a basal rate of 9 ATP/min and stimulated rate of ∼19 ATP/min at pH 8.5 (2X stimulation; not shown in manuscript). The stimulated rates at each pH also agree well with the basal rate of TRIP13 W221A (11 ATP/min at pH 7.5, 19 ATP/min at pH 8.5). We interpret these results as indicating that at pH 8.5, even without p31(comet):MAD2, PCH-2/TRIP13 is working closer to its maximum capacity than at pH 7.5, so the observed stimulation is lower as a result. We feel that the basal and stimulated ATPase rates measured at pH 7.5 are more relevant to the cellular context; however when initially characterizing these enzymes’ ATPase rates and *K*_*m*_/*k*_*cat*_ values, we sought to maximize the enzyme’s basal hydrolysis rate for easier characterization.

As noted above, we have added a new section to the Methods describing the conditions for our size exclusion analyses. This analysis was indeed done at pH 7.5, close to the conditions used for ATPase stimulation of PCH-2/TRIP13.

To address the point about hexamer stabilization and its effects on ATPase stimulation, we note two points: 1) While the TRIP13 hexamer is unstable, PCH-2 is a stable hexamer in all conditions, and we see the same general trend in pH-dependent changes in stimulation for PCH-2 as TRIP13 (for PCH-2, 1.7-fold stimulation at pH 7.5 versus 1.2-fold stimulation at pH 8.5; not shown in manuscript). 2) As mentioned above, we tested the concentration dependence of ATP hydrolysis by TRIP13 at both pH 7.5 and 8.5, and the results indicate that in our ATPase assay conditions, TRIP13 is likely to be fully hexameric even in the absence of p31(comet):MAD2.

*7) The authors show that only one p31/MAD2 complex binds to the TRIP13 hexamer, despite the presence of 6 N-terminal domains. A substrate delivery model is suggested in which p31 transiently interacts with the TRIP13 N-terminal domain before MAD2 directly binds in the central pore. However, currently this is largely speculation, and the authors should test this model by assessing the complex formation for the W221A pore loop mutant of TRIP13*.

This is an excellent suggestion, and was indeed an experiment that we initially attempted. For these experiments, we have used the TRIP13^E253Q^ mutant, which forms a stable hexamer by size exclusion. We generated the TRIP13^E253Q/W221A^ double mutant in order to test the role of the pore loop in p31(comet):MAD2 binding but found that, unexpectedly, this mutant does not form a stable hexamer like the TRIP13^E253Q^ mutant (as judged by elution on a size exclusion column in the presence of nucleotide). Instead, its behavior appears more similar to both wild-type TRIP13 and TRIP13^W221A^, which both adopt a mixture of oligomeric states whose migration on a size-exclusion column is not clearly distinct from that of p31(comet):MAD2. Thus, we were unable to accurately judge complex formation in the TRIP13^E253Q/W221A^ double mutant using elution-volume shifts on a size exclusion column. We have now noted this result in the text at the end of the subsection entitled “p31(comet) functions as an adapter between MAD2 and TRIP13”.

The behavior of the TRIP13^E253Q/W221A^ mutant is somewhat mysterious, and a full explanation will likely require a more comprehensive analysis of the *M. musculus* TRIP13^E253Q^ hexamer’s conformation. We note that the TRIP13^W221A^ single mutant has comparable activity to fully-stimulated TRIP13, strongly supporting the notion that it can form active hexamers. Also, *C. elegans* PCH-2^W221A^ forms stable hexamers as in wild-type PCH-2, indicating that this mutant does not significantly disrupt PCH-2 structure. We are currently pursuing several avenues for structural analysis of TRIP13, which may explain how the W221A mutation can disrupt the stable hexamer state adopted by TRIP13^E253Q^.

*8) It is described in Materials and methods that* “*…optimal basal ATPase rates were obtained from protein treated during the Ni*^*2+*^*-affinity purification step with 0.8 M urea, which removes the two ADP molecules bound to each hexamer…*”.

*It is unclear to me why the co-purified ADP needed to be removed prior to ATPase measurements, during which ADP is constantly generated and quickly turned into ATP by the regeneration system. Is this stripping step necessary to more accurately measure the PCH-2 concentration by UV absorbance? Or does PCH-2 with 2 bound ADP represent a trapped, inactive form of the hexamer, which may also be prevented by purifying PCH-2 in the presence of ATP*?

Based on our experience with PCH-2, we believe that the reviewer is most likely correct in labeling the ADP/Apo state we crystallized as a “trapped, inactive form of the hexamer.” This has stymied crystallization of PCH-2 with other nucleotides (as noted above), as well as initial ATPase activity assays. We found that the urea treatment gave maximally active PCH-2, likely because the ADP/Apo state is remarkably stable; the two ADP molecules imaged in our crystal structure remained bound to PCH-2 during a two-day protein purification (including four chromatography steps and multiple buffer exchanges) and several days in the crystallization conditions. It is likely that incubation in an ATP-regenerating system would eventually give maximally active protein, but the amount of time required for this activation would be prohibitive.

As can be observed in Figure 10, the A_280_ of purified PCH-2 changes very little after stripping out the bound nucleotides; however, it is clear from changes in A_260/280_ ratio that the treatment does indeed have the desired effect.

Figure 10 describes the stripping of nucleotide from PCH-2^E253Q^, intended for crystallization trials. The results for wild-type PCH-2 are equivalent.

Author response image 2.UV Absorbance Spectra of PCH-2.The denatured UV absorbance spectrum of full-length PCH-2^E253Q^ was collected after standard purification from *E. coli* overexpression (red line), or purification with a transient 0.8 M urea wash followed by stabilization of the protein in buffer containing 50 mM ammonium sulfate (blue line). The A_260/280_ ratios of the urea-treated sample agreed with the theoretical value calculated for PCH-2 protein based on amino acid content (0.689 measured versus 0.701 theoretical), and the ratio for untreated PCH-2 matched the theoretical value calculated for a PCH-2 hexamer bound to two ADP molecules (0.825 measured versus 0.818 theoretical), as observed in our crystal structure of untreated wild-type protein. We normalized the two spectra in the 295-300 nm range, which is not affected by bound nucleotides, then subtracted them to obtain a residual spectrum (green), which closely matched a scaled spectrum of ADP (black circles). Quantitation of ADP amounts from the residual spectrum using the molar absorbance of ADP (15,400 M^-1^cm^-1^ at 260 nm) and PCH-2 (38,400 M^-1^cm^-1^ at 280 nm) indicated that urea treatment removes ∼2.3 ADP molecules per PCH-2 hexamer.**DOI:**
http://dx.doi.org/10.7554/eLife.07367.023

*9) The ATPase measurements for the W221A mutant of TRIP13 showed absolutely no response to the presence of p31 and MAD2 at 27°C (*Figure 5*), but at 37°C there is a more than 30% increase in hydrolysis activity (*Figure 6—figure supplement 1*). The authors should try to explain those temperature-dependent differences in behavior*.

This is an interesting point, and one that we cannot readily explain. It is possible that the W221A mutant is not entirely defective in substrate recognition and engagement, and that the mutation’s effect varies somewhat at different temperatures. Even at 37°C, the mutant has much higher basal ATPase activity than wild-type TRIP13. Also, in our MAD2 conversion assays—performed at 37°C—showed no activity for TRIP13^W221A^ (Figure 7, bottom panel), indicating that even if this mutant is somewhat stimulated by p31(comet):MAD2 at 37°C, it is still unable to mediate C-MAD2 to O-MAD2 conversion.

*10) In the Discussion section the authors discuss TRIP13's mode of substrate recognition, which seems to include an initial interaction of the p31 adaptor with the TRIP13 N-terminal domain, before the MAD2 substrate binds in the central pore. The authors state:* “*its mode of substrate recognition, which parallels not ClpX but a family of ‘classic remodelers’ including NSF, p97, and Pex1.*” *I am sure the Corbett group is aware that many ClpX substrates with N-terminal degradation signals initially interact with the ClpX N-terminal domain before they are transferred to the central pore, and ClpX adaptors also bind to the N-terminal domain for substrate delivery to the central pore. The overall process of substrate delivery to TRIP13 thus appears to strongly parallel the mechanisms of ClpX, even though there are differences in the architecture of the N-terminal domains. The authors should clarify or correct their statement to prevent confusion*.

We thank the reviewer for pointing this out, and have altered the sentence in question to read: “…its mode of substrate recognition, which is mediated by an N-terminal domain related to a family of ‘classic remodelers’ including NSF, p97, and PEX1.”

*11) There is a mistake in a figure reference in the last paragraph of the subsection headed “p31(comet) functions as an adapter between MAD2 and TRIP13*”*. The sentence* “*Semi-quantitative analysis of Coomassie-stained gels, and light-scattering based molecular-weight measurements on this peak, revealed that for each TRIP13E235Q hexamer, about 1 copy of each protein was shifted into the TRIP13 peak (*Figure 4*, Figure 4–figure supplement 4)…*” *should reference*
Figure 4*, not 4D*.

Because of the overhaul of figure format and numbering, this figure panel has been changed (now Figure 5), but we have made sure that the text now refers to the correct figure panel.